# The acidic intrinsically disordered region of the inflammatory mediator HMGB1 mediates fuzzy interactions with CXCL12

Malisa Vittoria Mantonico [1,2], Federica De Leo[1,9], Giacomo Quilici[1], Liam Sean Colley [3,4], Francesco De Marchis[2,5], Massimo Crippa [5], Rosanna Mezzapelle [2,5], Tim Schulte[6], Chiara Zucchelli[1], Chiara Pastorello [1], Camilla Carmeno [1], Francesca Caprioglio [2,5], Stefano Ricagno[6,7], Gabriele Giachin [8], Michela Ghitti [1] ✉, Marco Emilio Bianchi [2,5] & Giovanna Musco [1] ✉

Chemokine heterodimers activate or dampen their cognate receptors during inflammation. The CXCL12 chemokine forms with the fully reduced (fr) alarmin HMGB1 a physiologically relevant heterocomplex (frHMGB1•CXCL12) that synergically promotes the inflammatory response elicited by the G-protein coupled receptor CXCR4. The molecular details of complex formation were still elusive. Here we show by an integrated structural approach that frHMGB1•CXCL12 is a fuzzy heterocomplex. Unlike previous assumptions, frHMGB1 and CXCL12 form a dynamic equimolar assembly, with structured and unstructured frHMGB1 regions recognizing the CXCL12 dimerization surface. We uncover an unexpected role of the acidic intrinsically disordered region (IDR) of HMGB1 in heterocomplex formation and its binding to CXCR4 on the cell surface. Our work shows that the interaction of frHMGB1 with CXCL12 diverges from the classical rigid heterophilic chemokines dimerization. Simultaneous interference with multiple interactions within frHMGB1•CXCL12 might offer pharmacological strategies against inflammatory conditions.

Chemokines constitute a large family of signaling proteins that interact with cell surface chemokine G protein-coupled receptors (GPCRs). Through an intricate network of cross-talks with their receptors they regulate leukocyte activation and trafficking in physiological and pathological conditions[1,2]. Chemokines are structurally characterized by the presence of two conserved disulfide bridges, a flexible N-terminus, a three-stranded β-sheet packing on a C-terminal α-helix and

3 loops connecting the elements of secondary structure. They usually exist in a monomer-multimer equilibrium. In particular, members of the CXC subfamily, like CXCL12 (C-X-C motif ligand 12), can dimerize via intermonomer contacts between the first β-strand and the α-helix of each monomer (Fig. 1a, Supplementary Fig. 1a). A shift towards one or the other form can either activate or dampen their cognate receptors, thus adding a further layer of complexity to the functional tuning

[1]Biomolecular NMR Laboratory, Division of Genetics and Cell Biology, IRCCS Ospedale San Raffaele, Milan, Italy. [2]School of Medicine, Università Vita e Salute-San Raffaele, Milan, Italy. [3]HMGBiotech S.r.l., 20133 Milan, Italy. [4]School of Medicine and Surgery, Università Milano-Bicocca, 20126 Milan, Italy. [5]Chromatin Dynamics Unit, Division of Genetics and Cell Biology, IRCCS Ospedale San Raffaele, Milan, Italy. [6]Institute of Molecular and Translational Cardiology, IRCCS Policlinico San Donato, Milan, Italy. [7]Department of Biosciences, Università degli Studi di Milano, Milan, Italy. [8]Department of Chemical Sciences (DiSC), University of Padua, 35131 Padova, Italy. [9]Present address: Experimental Therapeutics Program, IFOM ETS - The AIRC Institute of Molecular Oncology and AIRC, Fondazione AIRC per la Ricerca sul Cancro ETS, Milan, Italy. ✉e-mail: ghitti.michela@hsr.it; musco.giovanna@hsr.it

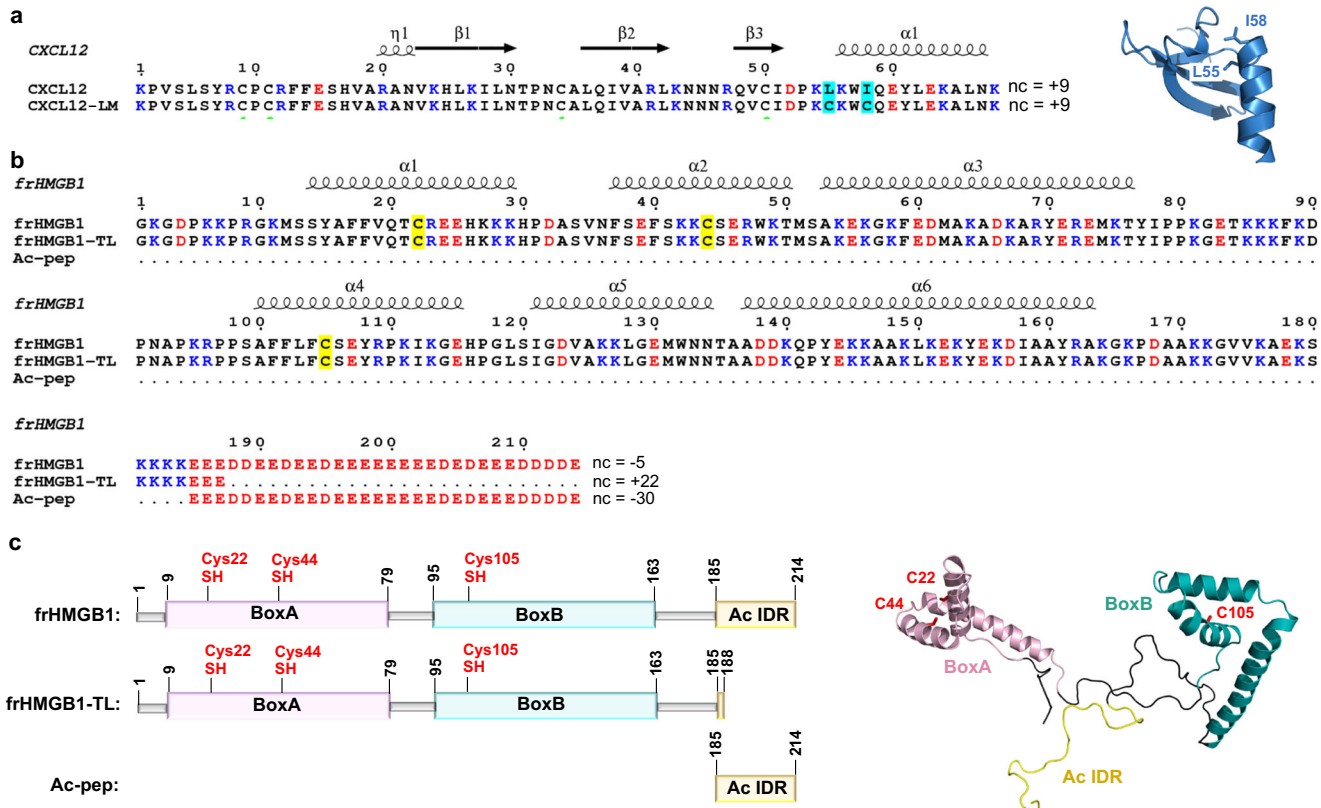

**Fig. 1 | CXCL12 and frHMGB1 constructs.** Amino acid sequence of: (**a**) CXCL12 and locked monomer CXCL12 mutant L55C/I58C (CXCL12-LM) and (**b**) frHMGB1, frHMGB1-TL and Ac-pep. Basic and acidic residues are shown in blue and red, respectively, nc indicates the net charge. L55C and I58C are colored in cyan on CXCL12 sequences. C22, C44 and C105 are colored in yellow on HMGB1 sequences. On the top of the alignments the elements of secondary structures are indicated. In (**a**) on the right is reported the cartoon representation of CXCL12 (pdb code: 2KEE), with L55 and I58 explicitly shown in sticks. **c** Schematic diagram of the fully reduced form of HMGB1 (frHMGB1), tail-less frHMGB1 (frHMGB1-TL), the acidic peptide (Ac-pep) corresponding to the HMGB1 acidic intrinsically disordered region (Ac IDR). BoxA (pink), BoxB (cyan) and Ac IDR (yellow) are represented with boxes and colored on frHMGB1 structure (AF-P63159). The side chains of fully reduced cysteines are represented in red sticks.

of chemokine/receptor axis and of their downstream pathways[3–5]. Another sophisticated mechanism of chemokine fine-regulation is their ability to heterodimerize with other chemokines, resulting in synergic activation and dimerization of their cognate receptors[6–8]. This mechanism is particularly relevant in inflammatory conditions, where chemokine heterocomplexes activate chemokine receptors in the presence of low concentrations of chemokine-selective agonists that otherwise, without the synergy-inducing partner, would be inactive[8,9]. Examples include the interaction of: (i) CCL21 or CCL19 with CXCL13 to enhance leukocyte migration and activities through binding and activation of CCR7 at lower agonist concentrations[10], (ii) CCL5 with CXCL4 to recruit monocytes and neutrophils[11], and (iii) CXCL12 and CXCL9 with CXCR4 to attract lymphoma cells[12]. Such a chemokine interactome further enlarges the possibility of heteromeric interactions among chemokines, expanding the fine-regulation of signaling possibilities[11,13].

Intriguingly, chemokines can also form heterophilic interactions with some inflammatory mediators that are not structurally homologous to the classical CC-, CXC-, CX3C-, or XC-chemokines[4]. In this sense CXCL12 represents a paradigmatic example, as it is also able to bind to other proteins, such as galectins[14] and the alarmin High Mobility Group Box 1 (HMGB1)[6,15]. The interaction with Galectin 3 is immunoregulatory and attenuates CXCL12-stimulated signaling via CXCR4[14]. Conversely, binding of CXCL12 to HMGB1 synergistically enhances the CXCR4-dependent chemotactic response of monocytes and is involved in tissue regeneration, cell proliferation and tumour progression[6,15,16]. The 25 kDa HMGB1 protein comprises two L-shaped HMG tandem boxes (~80 aa each connected by a flexible linker),

referred to as BoxA and BoxB, and an acidic C-terminal intrinsically disordered region (IDR, 30 aa) (Fig. 1b, c). HMGB1 is a Damage Associated Molecular Pattern (DAMP), which once released in the extracellular space alerts the host to stress, unscheduled cell death or microbial invasion, thus triggering inflammation and immune responses[17,18]. HMGB1 is passively released by dead non-apoptotic cells and is actively released by severely stressed cells and by immune cells such as macrophages, natural killer cells, neutrophils and mature dendritic cells (reviewed in[19]). HMGB1 contains three cysteines (C22, C44, and C105) (Fig. 1b, c), whose redox states determine how it functions as a pro-inflammatory mediator[20]. On one hand, the disulfide form (dsHMGB1), with C22 and C44 forming a disulfide bridge on BoxA, binds to the Toll-like receptor 4/MD2 complex, herewith promoting inflammatory responses and cytokine activation[21]. On the other hand, the fully reduced form (frHMGB1) plays a pivotal role in promoting the recruitment of inflammatory cells to injured tissues via heterocomplex formation with CXCL12 (frHMGB1•CXCL12) and activation of CXCR4. This heterocomplex induces specific CXCR4 homodimer rearrangements, promotes CXCR4-mediated signaling, resulting in increased ERK activation and calcium rise induction[15] and maintains CXCR4 on the plasma membrane in a β-arrestin 2 dependent manner[22]. HMGB1, CXCL12 and CXCR4 are highly expressed in inflammation related cancers, like malignant mesothelioma[23], where they contribute to disease initiation and progression[16]. Pharmacological targeting of the frHMGB1•CXCL12•CXCR4 axis is thus emerging as an appealing opportunity against inflammation related diseases[24].

While Galectin 3 is structurally reminiscent of chemokines and interacts with CXCL12 exploiting in part the CXC-type dimerization

surface, composed by the beta-strand β1 and the alpha-helix α1[14] (Fig. 1a, Supplementary Fig. 1a), the molecular details dictating frHMGB1•CXCL12 intermolecular interactions are in part elusive, possibly because of the intrinsic dynamics of the system components. On one side CXCL12, as a classical chemokine, exists in a physiologically relevant monomer-dimer equilibrium[25]. On the other side, HMGB1 is distinguished by conformational heterogeneity and variable contact patterns: it oscillates between a collapsed and open form, via intramolecular electrostatic interaction between the acidic IDR and the basic HMG boxes[26–29]. Whether one or two molecules of CXCL12 bind to HMGB1 and whether the acidic IDR plays a role in heterocomplex formation and function are still open questions[5,30].

Here, to address these issues, we adopt a dissecting and mutagenesis strategy coupled to an integrative structural approach (NMR, AUC, ITC, MST, SAXS). Our findings uncover the formation of an equimolar complex between frHMGB1 and CXCL12, with the acidic IDR assuming a previously overlooked, yet crucial role in complex formation. The resulting complex is best described by a heterogeneous ensemble rather than by a single defined structure, with frHMGB1 retaining its intrinsic dynamics during its interaction with CXCL12. In particular, NMR titrations and [15]N relaxation experiments indicate that the acidic IDR contains multiple sites capable of interacting with CXCL12 and maintains its conformational flexibility when bound to CXCL12. Moreover, CXCL12 employs the same interaction surface to establish multivalent interactions with both BoxA and the acidic IDR. Importantly, we provide evidence that in a cellular context the acidic IDR facilitates the binding of the frHMGB1•CXCL12 heterocomplex to CXCR4 on the extracellular surface. All these features align well with the concept of fuzzy interactions, where at least one component retains its dynamic nature and has a discernible impact on functional outcomes[31,32].

## Results

### HMGB1 acidic IDR takes part in frHMGB1•CXCL12 formation
The molecular details and the actual role of the single HMGB1 domains in the formation of the heterocomplex with CXCL12 are still elusive. Existing models, based on NMR titrations between the single HMG boxes and CXCL12, are based on the assumption that only the structured domains are the main actors in complex formation[30]. Whether the acidic IDR of frHMGB1 plays a role in complex formation has never been explored. We have therefore adopted a dissecting approach and performed comparative nuclear magnetic resonance (NMR) experiments titrating [15]N CXCL12 with a synthetic peptide corresponding to the acidic IDR (Ac-pep), a tail-less HMGB1 construct, composed of the fully reduced HMG tandem domain devoid of the acidic IDR (frHMGB1-TL), and a fully reduced full-length HMGB1 (frHMGB1) (Fig. 1b, c).

Notably, addition of Ac-pep to [15]N CXCL12 induced a dramatic change of the corresponding [1]H-[15]N Heteronuclear Single Quantum Coherence (HSQC) spectrum, with substantial chemical shift perturbations (CSPs, in the 0.2–0.6 ppm range) and overall peak intensity reduction (Fig. 2a–c). The interaction occurred in the intermediate exchange on the NMR chemical shift time scale: peaks were severely broadened or disappeared beyond detection upon addition of sub-stoichiometric amounts of Ac-pep, but reappeared at 1:1 stoichiometry (Supplementary Fig. 2a). The dissociation constant was in the sub-micromolar range ($K_d = 0.2 \pm 0.1\,\mu M$), as assessed by line-shape analysis using the software TITAN 1.6[33] (Supplementary Fig. 2b). CXCL12 residues with the highest CSPs were R12, H17, V18, R20, V23, K24, H25, A40, R41, N45, W57, L66, N67 (Fig. 2b), with V23, K24, H25 (on β1) and L66, N67 (on α1) all located on the known CXCL12 homodimerization interface (Fig. 2d, Supplementary Fig. 1a).

In contrast to Ac-pep, addition of frHMGB1-TL to [15]N CXCL12 (Fig. 2e–g) induced very small CSPs (in the 0.02–0.06 ppm range). As typically observed in NMR studies of chemokine heterocomplex formation, the interaction occurred again in the intermediate exchange

regime on the NMR chemical shift timescale[14], with peak-intensity reduction upon binding. Residues mostly affected by the interaction with frHMGB1-TL partially coincided with the ones affected by Ac-pep (H17, V23-H25, A40, N45, N67), suggesting that both the acidic IDR and the HMG tandem domain in part share the same interaction surface (Fig. 2h). Similarly, titration of full-length frHMGB1 induced significant CSPs (in the 0.05–0.1 ppm range) on residues located on the β1 strand (V23-H25) and the α-helix (L66, N67) (Fig. 2I, j). Herein, we observed pronounced line broadening effects already at sub-stoichiometric concentrations (1:0.5) (Fig. 2i) that hampered analysis at equimolar ratio, suggesting different binding dynamics and affinities for the full-length protein and its different constructs. Of note, comparison of the profiles of the CSPs of [15]N CXCL12 upon addition of either Ac-pep, frHMGB1-TL or frHMGB1 were similar, with residues located on the CXCL12 dimerization surface showing the highest CSPs (Fig. 2l). Importantly, the chemical shifts of these residues and their perturbations strongly depend on the CXCL12 monomer-dimer equilibrium[5]. Thus, to distinguish CSPs due to direct interactions with frHMGB1 and constructs thereof from those related to potential changes in the CXCL12 oligomerization state, we repeated the titrations with a CXCL12 mutant locked in a monomeric state (CXCL12-LM)[34]. This mutant, which restricts the α1 region to a specific orientation relative to the β-sheet through a disulfide bond (between C55 and C58) (Fig. 1a, Supplementary Fig. 1b), was previously engineered with the aim to block chemokine oligomerization and to distinguish CSPs resulting from ligand induced dimerization from those stemming from direct interactions[34,35]. Upon the addition of Ac-pep, frHMGB1-TL, and frHMGB1, we observed that the spectral perturbations in CXCL12-LM were similar to those seen in the wild-type CXCL12 in terms of line broadening and CSP profiles. However, the CSPs magnitude of [15]N CXCL12-LM upon addition of frHMGB1-TL and frHMGB1 was larger than that observed in the wild-type protein, possibly due to different internal dynamics and/or binding kinetics in the wild-type and mutant protein. Nonetheless, the resemblance in CSP profiles strongly suggests that the N-terminal portion of the β1-strand and the α1-helix of CXCL12 indeed constitute an authentic interaction surface (Supplementary Fig. 3).

Taken together, comparison of [15]N CXCL12 NMR titrations with frHMGB1 fragments revealed that the acidic IDR of HMGB1 is directly involved in frHMGB1•CXCL12 heterocomplex formation and that the CXCL12 dimerization surface works as hub for multivalent interactions with both the acidic IDR and the HMG tandem domain.

### The frHMGB1•CXCL12 heterocomplex forms via fuzzy interactions
We next hypothesized that the inter-molecular interactions between the acidic IDR and CXCL12 might perturb frHMGB1 intra-molecular interactions and conformational equilibria. Indeed, in reversed titrations, frHMGB1 amide resonances of spy residues reported to interact with the acidic IDR (e.g. W48, T76, I78 on BoxA, and A93, I158 on BoxB)[28], moved towards their NMR frequency in the tailless construct upon addition of CXCL12. These displacements suggest a weakening of the intramolecular interactions between the acidic IDR and the frHMG boxes (Fig. 3a). The shifts towards frHMGB1-TL resonances were relatively small, presumably because CXCL12 only partially competes with frHMGB1 intra-molecular interactions. Addition of an equimolar amount of Ac-pep to [15]N frHMGB1 in complex with CXCL12 was sufficient to sequester CXCL12 and to disrupt the heterocomplex, as indicated by the reappearance of [15]N frHMGB1 resonances, confirming the important contribution of the acidic IDR to heterocomplex formation (Fig. 3b).

Overall, CSPs and intensity variations in [15]N frHMGB1/CXCL12 NMR titrations are due to both intra- and inter-molecular interactions, hence mapping of spectral perturbations on the frHMGB1 structure reflects both phenomena, which are difficult to separate (Fig. 3c–e).

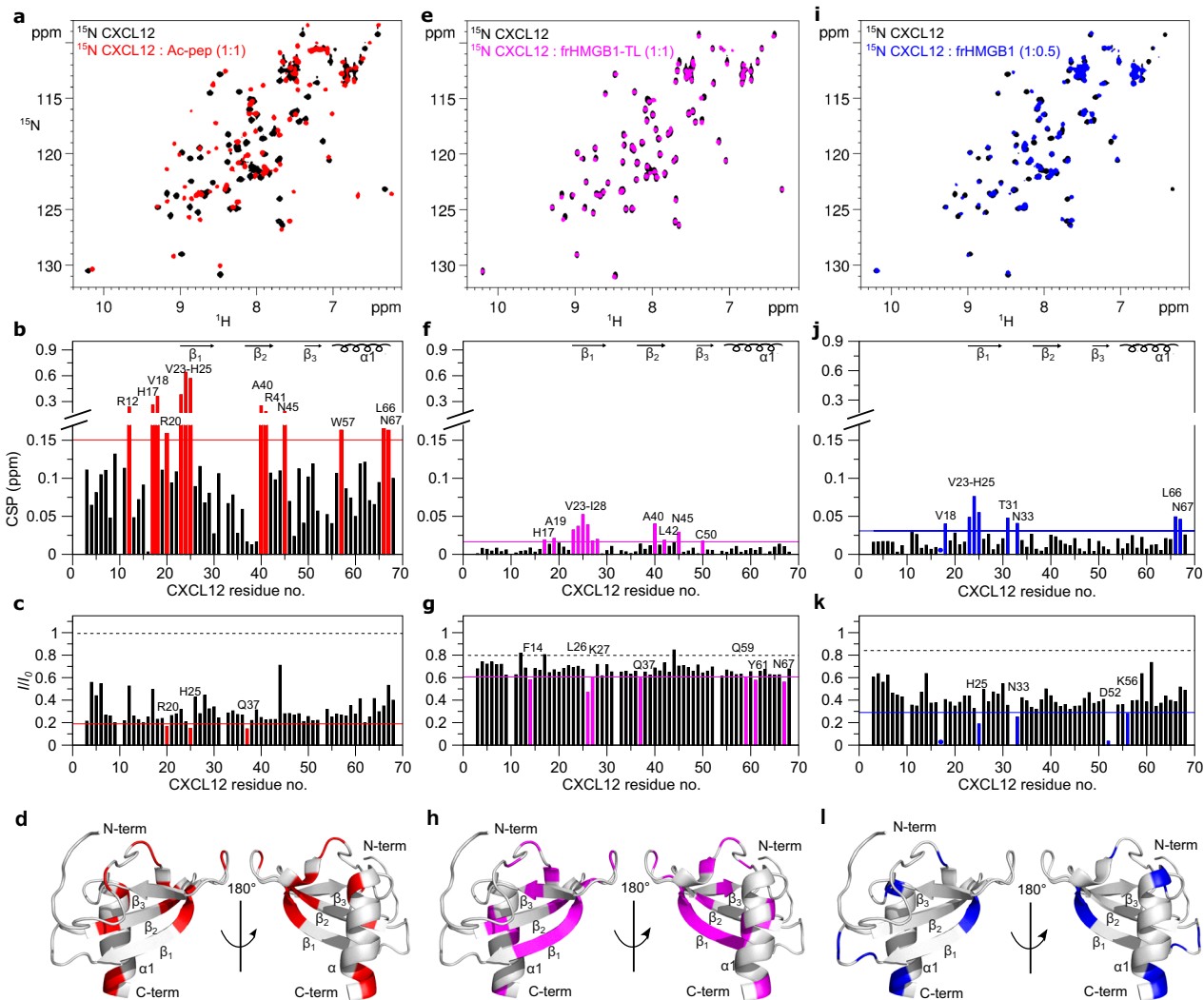

**Fig. 2 | Ac-pep, frHMGB1-TL and frHMGB1 interact with CXCL12 dimerization surface. a** Superposition of ¹H-¹⁵N HSQC spectra of ¹⁵N CXCL12 (0.1 mM) without (black) and with (red) Ac-pep (1:1). Bar graph showing (**b**) residue-specific chemical shift perturbation (CSPs) and **c** peak intensities ratios (*I/I₀*) of ¹⁵N-labeled CXCL12 (0.1 mM) upon addition of Ac-pep (1:1). Residues with CSP > avg + σ₀ (corrected standard deviation, red line) and with *I/I₀* <avg - SD (standard deviation, red line) are labeled and (**d**) shown in red on CXCL12 (gray cartoon, pdb code: 2KEE). **e** Superposition of ¹H-¹⁵N HSQC spectra of ¹⁵N CXCL12 (0.1 mM) without (black) and with (magenta) frHMGB1-TL (1:1). Bar graph showing (**f**) residue-specific CSPs and (**g**) I/I₀ of ¹⁵N-labeled CXCL12 (1:1) of ¹⁵N-labeled CXCL12 (0.1 mM). Residues with CSP > avg + σ₀ (magenta line) with *I/I₀*

<avg - SD (magenta line) are labeled and (**h**) shown in magenta on CXCL12. **i** Superposition of ¹H-¹⁵N HSQC spectra of ¹⁵N CXCL12 (0.1 mM) without (black) and with (blue) frHMGB1 (1:0.5). Bar graph showing (**j**) residue-specific CSPs and (**k**) I/I₀ of ¹⁵N-labeled CXCL12 (0.1 mM) upon addition of frHMGB1 (1:0.5). Residues with CSP > avg + σ₀ (blue line) and *I/I₀* < avg - SD (blue line) are labeled and (**l**) shown in blue on CXCL12. In the bar graphs α-helices and β-strands are schematically represented on the top, missing residues are prolines, dots indicate residues disappearing upon binding, the dashed black line indicates the expected peak intensity decrease due to the titration dilution effect. Source data are provided as a Source Data file.

Thus, to remove the confounding effect of the acidic IDR, we performed NMR titrations of ¹⁵N frHMGB1-TL with CXCL12. Notably, the CSPs and peak intensity reduction profiles were different and substantially smaller than the ones observed in the full-length protein. Nonetheless, the removal of the acidic IDR brought out the presence of an interaction surface formed by the first two helices of BoxA, with no major involvement of BoxB (Fig. 3f–h). The relevant role of BoxA in the interaction was further emphasized by the fact that addition of CXCL12 to the oxidized tailless construct (¹⁵N dsHMGB1-TL), produced markedly reduced spectral changes (Supplementary Fig. 4a) compared to frHMGB1-TL (Fig. 3f, g). A similar result was obtained in the reverse titration (Supplementary Fig. 4b). This confirms that BoxA and its oxidation state are indeed crucial for the interaction with CXCL12 (ref. 20). Notably, and similar to frHMGB1, in NMR titrations of ¹⁵N dsHMGB1 and CXCL12 (Supplementary Fig. 4c) and reverse (Supplementary Fig. 4d), the major spectral perturbations were mainly due to

the intra- and inter-molecular interactions of the acidic IDR, whose binding to CXCL12 is independent from the redox state of BoxA.

The dynamic nature of the acidic IDR interaction with CXCL12 was further confirmed by NMR titrations of recombinant ¹⁵N acidic IDR (Ac-pep_rec) with unlabelled CXCL12. As is typical for fuzzy interactions involving IDRs or intrinsically disordered proteins (IDPs)[36,37], the ¹H-¹⁵N HSQC spectrum of ¹⁵N Ac-pep_rec, in both the absence and presence of equimolar CXCL12, displayed reduced peak dispersion and high signal overlap, with minimal chemical shift perturbations (Fig. 4a). Similarly, Hα-Cα and Hβ-Cβ peak clusters of aspartic and glutamic residues in the ¹H-¹³C HSQC spectra did not show significant chemical shift changes upon CXCL12 binding, indicating the absence of persistent secondary structure formation (Fig. 4b). The pronounced overlap in the NMR spectra and the repetitive amino acid sequence prevented specific residue assignments. As a result, peaks were assigned arbitrary numbers, and the interpretation of their intensities should primarily be

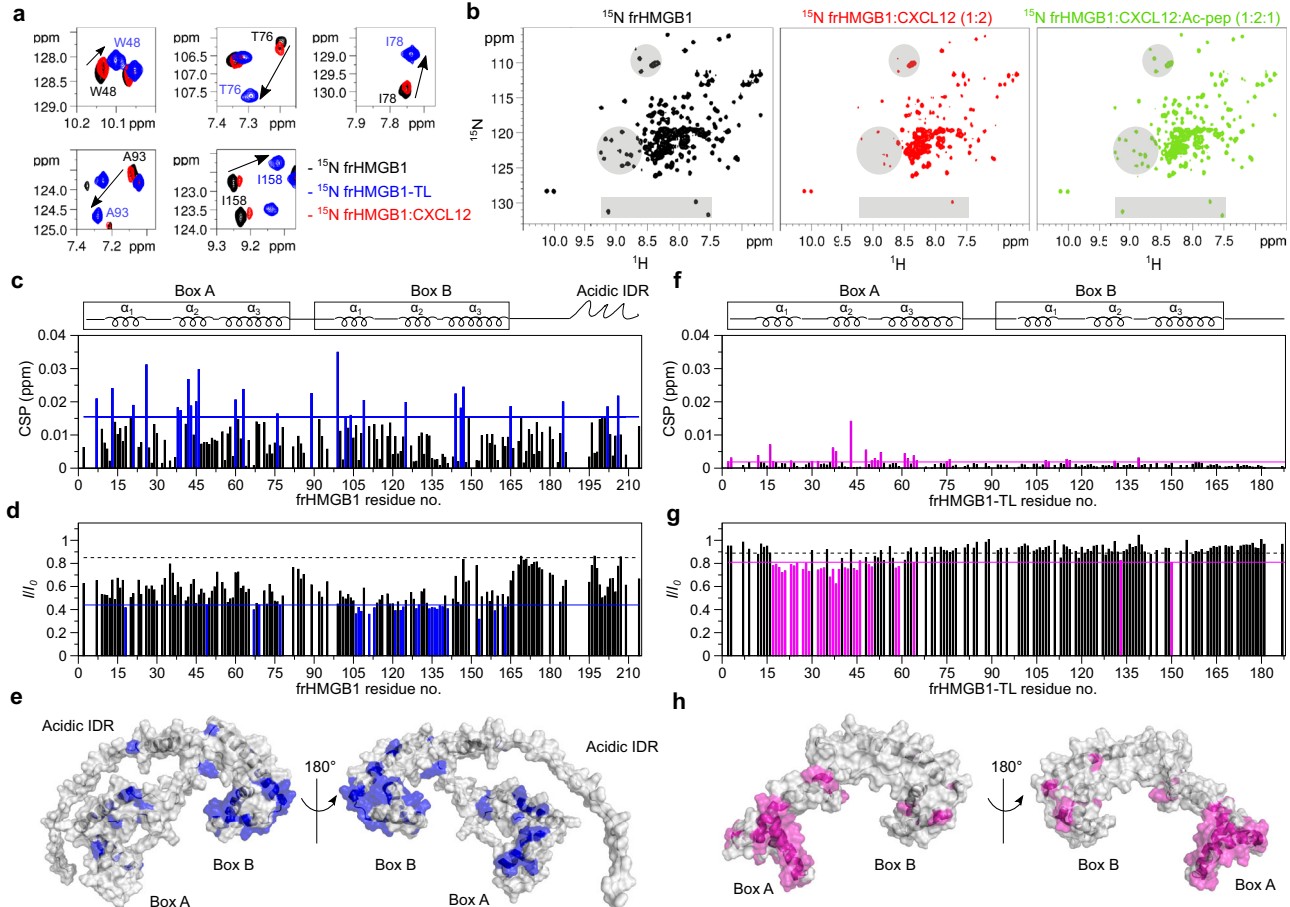

**Fig. 3 | The frHMGB1•CXCL12 heterocomplex forms via fuzzy interactions.**
**a** Superposition of selected regions of ¹H-¹⁵N HSQC spectra of 0.1 mM ¹⁵N frHMGB1 (corresponding to spy residues W48, T76, I78, A93, I158) without (black) and with 0.2 mM CXCL12 (red) and 0.1 mM ¹⁵N frHMGB1-TL (blue). CXCL12 partially competes with intramolecular frHMGB1 interactions and specific amide resonances move (arrow) towards the chemical shift of the corresponding amide in the tailless construct. **b** ¹H-¹⁵N HSQC spectra of frHMGB1 (0.1 mM) without (black) and with 0.2 mM CXCL12 (red), and with subsequent addition of 0.1 mM Ac-pep (green). Grey shadowed regions highlight resonances disappearing and reappearing upon addition of CXCL12 and Ac-pep, respectively. Bar graphs showing (**c**) residue-specific chemical shift perturbation (CSP) and (**d**) peak intensity ratios ($I/I_0$) of ¹⁵N-labeled frHMGB (0.1 mM) upon addition of CXCL12 (1:1). Residues with

CSP > avg + $\sigma_0$ (corrected standard deviation, blue line) and $I/I_0$ <avg - SD (standard deviation, blue line) are colored (blue) and (**e**) mapped on frHMGB1 (grey surface, Aphafold2 model AF- P63159). Bar graph showing (**f**) residue-specific CSPs and (**g**) $I/I_0$ of ¹⁵N-labeled frHMGB1-TL (0.1 mM) upon addition of CXCL12 (1:1). Residues with CSP > avg + $\sigma_0$ (magenta line) and $I/I_0$ <avg - SD (magenta line) are colored in magenta and (**h**) mapped on frHMGB1-TL (grey surface, pdb code: 2YRQ). In the bar-graphs α-helices are schematically represented on the top, missing residues are either prolines, or superimposed residues of the acidic IDR or absent because of exchange with the solvent, the dashed black line indicates the peak intensity decrease due to the titration dilution effect. Source data are provided as a Source Data file.

considered qualitative (Fig. 4c, d). Herein, analysis of ¹⁵N relaxation experiments (heteronuclear NOE, $R_1$, $R_2$) for free and bound ¹⁵N Ac-pep$_{rec}$ supported the formation of a fuzzy complex. Both states exhibited negative heteronuclear NOE values, signifying high flexibility on the picosecond to nanosecond timescale. A modest increase in complex heteronuclear NOE values indicated a slight reduction in peptide mobility in the presence of CXCL12. The increase in $R_2/R_1$ and reduced peak intensity was in line with complex formation, with the consequent slowdown of its tumbling in solution and with the dynamic exchange between multiple CXCL12 binding sites (Fig. 4e, f). The presence of multiple interchangeable binding sites within the acidic IDR was further confirmed by titrating ¹⁵N CXCL12 with peptide fragments of the IDR (Ac-pep$_{185-195}$ and Ac-pep$_{204-214}$), which similarly interacted with the CXCL12 dimerization surface (Supplementary Fig. 5a–f).

Taken together, the combination of the different NMR titrations and the relaxation data on ¹⁵N Ac-pep$_{rec}$ indicate that frHMGB1•CXCL12 is a fuzzy dynamic heterocomplex, characterized by multivalent inter- and intra-molecular equilibria involving CXCL12, the HMG tandem

domain and the acidic IDR as major players within this intricate network of interactions.

**The acidic IDR binds CXCL12 via long-range electrostatics**
We next adopted the same dissection approach to investigate the thermodynamics of CXCL12 interaction with HMGB1, using a combination of isothermal titration calorimetry (ITC), microscale thermophoresis (MST) and fluorescence measurements. ITC injection of CXCL12 into Ac-pep solution generated spikes with a biphasic profile, indicative of different binding events, and large maximal exothermic heat changes (~ −0.6 μcal/s). Global fitting of the buffer-subtracted binding isotherm yielded an apparent $K_{d1}$ of 0.6 ± 0.1 μM and $K_{d2}$ of 0.1 ± 0.1 μM, in agreement with the low micromolar affinity estimated by NMR line-shape analysis (Fig. 5a, Table 1)[38,39]. Also the interaction between frHMGB1 and CXCL12 appeared biphasic and exothermic (Fig. 5b), though with one order of magnitude reduced amplitude (~ −0.06 μcal/s). The fitting of the curve yielded an apparent $K_{d1}$ = 1.2 ± 0.4 μM and a second one, whose nanomolar value should be taken with caution because of the large error in the global fitting[39]

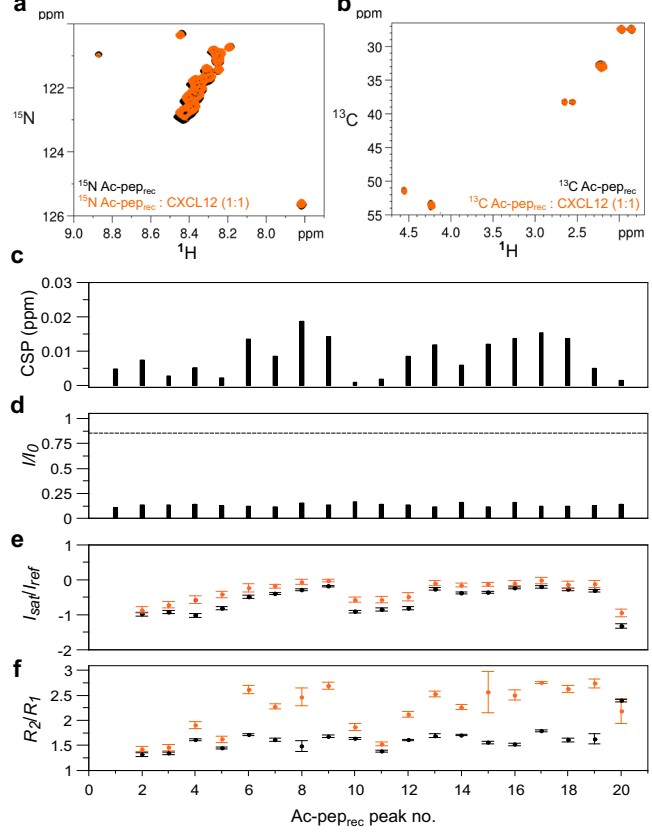

**Fig. 4 | The acidic IDR peptide ($^{15}$N Ac-pep$_{rec}$) dynamically interacts with CXCL12.** Superposition of (**a**) $^1$H-$^{15}$N HSQC spectra of $^{15}$N Ac-pep$_{rec}$ (0.1 mM) without (black) and with (orange) CXCL12 (1:1), (**b**) $^1$H-$^{15}$C HSQC spectra of $^{15}$N/$^{13}$C Ac-pep$_{rec}$ (0.1 mM) without (black) and with (orange) CXCL12 (1:1). Bar graphs showing (**c**) chemical shift perturbation (CSPs) and (**d**) peak intensity ratios ($I/I_0$) of $^{15}$N Ac-pep$_{rec}$ (0.1 mM) upon addition of CXCL12 (1:1). **e** Peak intensity ratios of heteronuclear NOE with and without proton saturation ($I_{sat}/I_{ref}$), error bars were calculated by error propagation from the standard deviation (SD) of the average value of the noise in the saturated $\langle I_{noise,sat} \rangle$ and non saturated reference $\langle I_{noise,ref} \rangle$ spectra. **f** Ratios of $R_2$ and $R_1$ relaxation rates of $^{15}$N Ac-pep$_{rec}$ (0.1 mM) without (black) and with CXCL12 (1:1) (orange). Data of one representative experiment performed on $n = 2$ biologically independent samples are presented as mean values +/− SD. Error bars are derived from relaxation data as described in methods. Amide resonances were not sequence specifically assigned and peaks were attributed arbitrary numbers. Source data are provided as a Source Data file.

(Table 1). Importantly, the low micromolar affinities measured by ITC for Ac-pep and frHMGB1 were in good agreement with the ones deriving from fluorescence and MST experiments, respectively (Fig. 5d, Table 1). While the heat of reaction between CXCL12 and frHMGB1-TL was not sufficient to derive any binding parameters (endothermic spikes at baseline level, -0.05 μcal/s) (Fig. 5c), a change of the thermophoretic diffusion properties of fluorescently labeled CXCL12 was detectable in the presence of frHMGB1-TL (Fig. 5d). Notably, the derived affinity ($K_d = 12.5 \pm 5.5$ μM) was almost one order of magnitude weaker than the one measured with full-length frHMGB1 ($K_d = 1.7 \pm 0.2$ μM), thus confirming the important role of the acidic IDR in heterocomplex formation (Table 1).

We reasoned that the interaction might be dominated by long range electrostatic interactions between the acidic IDR and the basic surface of CXCL12 (Supplementary Fig. 6a, b). Indeed, the heat of reaction associated to CXCL12 interaction with Ac-pep or frHMGB1 at higher ionic strength (150 mM NaCl) did not yield any spikes above baseline (Supplementary Fig. 6c, d). Also in fluorescence and MST

experiments binding affinities were reduced by one order of magnitude, confirming the prominent contributions of long-range electrostatic interactions (Fig. 5e, Table 1). As further confirmation that the acidic IDR mainly interacts with CXCL12 via long range electrostatic interactions, we titrated into $^{15}$N CXCL12 a negative control peptide corresponding to the arginine/lysine-rich tail of the HMGB1 mutant form leading to the brachyphalangy, polydactyly and tibial aplasia malformation syndrome[40]. As expected, no interaction was detected (Supplementary Fig. 5g–i). However, as already mentioned, the acidic IDR is not the unique driving force in the interaction, as the oxidation status of BoxA is also relevant for the binding to CXCL12, as indicated by the reduced affinity of dsHMGB ($K_d = 37.3 \pm 3.3$ μM) with respect to frHMGB1 (Supplementary Fig. 4e).

Collectively, ITC, MST, and fluorescence measurements, in accordance with NMR titrations, support a scenario in which the frHMG tandem domain and the acidic IDR both contribute to heterocomplex formation, with the acidic IDR working as an antenna for recruiting CXCL12 via long-range electrostatic interactions.

## frHMGB1 and CXCL12 form an equimolar heterocomplex

Previously, we and others postulated that frHMGB1 and CXCL12 interact with a 1:2 ratio[15,30]. To verify this hypothesis, we rigorously assessed complex stoichiometry by analytical ultracentrifugation (AUC). Initially, we compared sedimentation velocity AUC (SV-AUC) experiments of the individual and combined components at different stoichiometric ratios[41]. Sedimentation coefficient distributions ($c(s)$) of free CXCL12 (1.1 S) and frHMGB1 (2.3 S) were in good agreement with their apparent molecular weights (8.7 and 27 KDa, respectively) (Fig. 6a, b; Supplementary Table 1), and the corresponding frictional ratios ($f/f_o$) of 1.3 and 1.4 were in line with the globular and elongated shape of CXCL12 and frHMGB1, respectively (Supplementary Table 1). At variance to stable complexes, characterized by distinct SV-AUC curves for the bound and individual components, the frHMGB1•CXCL12 heterocomplex presented only two separable sedimentation distributions: one relatively sharp peak at lower $s$, corresponding to free CXCL12, and one at higher $s$ values, deriving from the sedimentation of free and bound frHMGB1. The latter, both at 50 mM (Fig. 6c, Supplementary Table 1) and 150 mM NaCl (Supplementary Fig. 7a–c, Supplementary Table 2), was relatively broad and shifted towards higher $s$ values (and apparent MWs) with increasing CXCL12 concentrations. This behavior is typically observed in highly dynamic complexes, where the reaction boundaries between bound and unbound species cannot be resolved within the signal-to-noise ratio of the experiment[42]. Importantly, analysis of SV-AUC curves of frHMGB1-TL with increasing concentrations of CXCL12 showed only two peaks corresponding to the free components, and no sedimentation peaks displacement or broadening were observed. Conceivably, without the acidic IDR the association between the two components is too rapid to be detected in the sedimentation time scale (Supplementary Fig. 7d–f).

Next, to determine the ratio of frHMGB1 and CXCL12 in the complex distribution, we performed multi-signal sedimentation velocity analytical ultracentrifugation experiments (MSSV-AUC) and exploited the different extinction coefficients of frHMGB1 and CXCL12 at minimal (250 nm) and maximal (280 nm) wavelengths to distinguish between the two complex components[43,44]. Analysis of MSSV-AUC suggested that frHMGB1 predominantly forms with CXCL12 (or CXCL12-LM) a 1:1 heterocomplex in solution, as indicated by the equimolar concentrations for both CXCL12 (CXCL12-LM) and frHMGB1 obtained from the peak area at ~3 S (Fig. 7d–f, Supplementary Table 1–3, Supplementary Fig. 7g–k).

Based on these experiments we conclude that frHMGB1 and CXCL12, in contrast to previous assumptions, form an equimolar heterocomplex, with the acidic IDR playing a fundamental role in complex assembly.

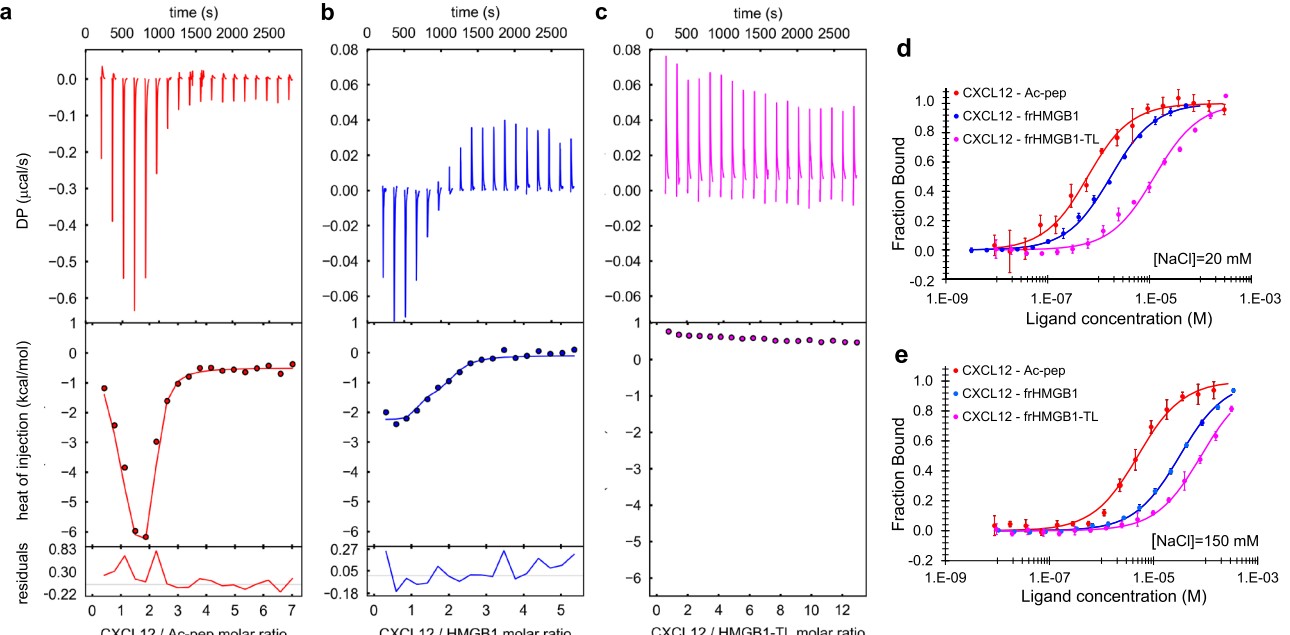

**Fig. 5 | The acidic IDR of frHMGB1 interacts with CXCL12 via long-range electrostatic interactions.** ITC measurements of CXCL12 titrated into (**a**) Ac-pep (red), (**b**) frHMGB1 (blue) and (**c**) frHMGB1-TL (magenta) (20 mM TrisHCl at pH 7.5, 50 mM NaCl). The upper, middle and lower panels show, respectively, the ITC sequential heat pulses (DP, differential power) for binding, the integrated data corrected for heat of dilution and the residuals. Data in (**a**, **b**) were globally fitted. Data in **c** could not be fitted because the heat of reaction was too small to be fitted with a nonlinear least-squares method. Data represents peak integration of ITC signal. One representative curve ($n = 2$) for each titration is shown. Normalized variation of fluorescence of 5,6-FAM-labelled Ac-pep upon addition of CXCL12 (red) and normalized variation of MST signal of labeled CXCL12 in the presence of frHMGB1 (blue) and of frHMGB1-TL (magenta), with (**d**) 20 mM NaCl and with (**e**) 150 mM NaCl. Data are presented as mean values +/− SD of $n = 3$ independent replicates. The thermodynamic parameters and $K_d$s are summarized in Table 1. Source data are provided as a Source Data file.

## SAXS supports the dynamic nature of frHMGB1•CXCL12

We then utilized small angle X-ray scattering (SAXS) to obtain low-resolution structural information on the estimated mass and shape of the heterocomplex. Primary analysis of the SAXS scattering curves of free CXCL12 and frHMGB1 yielded radii of gyration ($R_g$) of 1.52 nm and 2.60 nm, respectively, as well as pairwise distance distribution plot, P($r$), derived maximum distance ($D_{max}$) values of 4.8 nm and 8.5 nm, respectively. These values indicate that both proteins are monomeric and monodisperse in solution (Fig. 7a, Supplementary Table 4, Supplementary Fig. 8a, 9, 10). The values obtained for frHMGB1 are in accordance with a previous report[27]. The normalized Kratky representation of free CXCL12 and frHMGB1 displayed upward trends at higher $qR_g$ values, well in agreement with the presence of flexible regions within a folded core (Fig. 7b). While CXCL12 displayed a relatively symmetric P($r$) plot, the one of frHMGB1 had an asymmetric, elongated shape, tailed off to large distances (Supplementary Fig. 8b). This feature suggests the existence of multiple conformations in solution, and aligns with the presence of flexible regions and intramolecular multivalent fuzzy interactions between the acidic IDR and the basic HMG domains[27]. Thus, to generate a suitable ensemble of conformers able to account for HMGB1 intrinsic dynamics and match its experimental scattering curve, we used Ensemble Optimization Method (EOM)-based rigid body modelling[27,45] (Supplementary Fig. 11).

Analysis of the heterocomplex was challenging due to its transient and dynamic nature, which prevented its isolation through size exclusion chromatography and direct coupling to SAXS. Instead, we used a batch-mode strategy where frHMGB1 incubated with increasing CXCL12 concentrations was immediately analyzed by SAXS (Supplementary Fig. 12). In the SAXS spectrum of a protein complex the relative contribution of the individual components to the overall scattering pattern varies according to their size and scattering intensity[46]. Thus, the heterocomplex, which is the biggest component

(33.8 kDa), was expected to contribute most to the scattering signal and to dominate over the smallest one (CXCL12, 9.1 kDa). Indeed, in the presence of two equivalents of CXCL12, where more than 96% of HMGB1 was saturated with CXCL12, a 1:1 complex with an estimated molecular weight of 35 kDa was obtained (Supplementary Table 4). At higher CXCL12 concentrations, where a larger fraction of unbound CXCL12 was present, it was not possible to obtain a reliable estimate of the complex molecular weight (Supplementary Fig. 8a). We therefore focused our SAXS analysis on the sample comprising two CXCL12 equivalents. We observed an overall increase of the derived parameters with respect to free frHMGB1, compatible with complex formation ($R_g$ of 2.93 nm and $D_{max}$ of 10.5 nm) and a mass estimation corresponding to frHMGB1•CXCL12 heterocomplex (Supplementary Fig. 8a). The normalized Kratky plot and the P($r$) plot confirmed the dynamic and flexible nature of the complex, containing both folded domains and unstructured segments (Fig. 7b, Supplementary Fig. 8b).

Next, to model more quantitatively the fuzzy nature of frHMGB1•CXCL12 heterocomplex, we combined NMR-guided rigid-body docking (SASREF[47] and FoXSDock[48]) to EOM[45]. Since our NMR data indicated that CXCL12 can interact with both BoxA and the acidic IDR, two initial docking models were generated. To guide the interaction between CXCL12 on BoxA we used SASREF[47] taking advantage of the experimental CSPs (Supplementary Fig. 13a). To model the association between CXCL12 and the acidic IDR, we opted for FoXSDock[48] (Supplementary Fig. 13b), which is more suitable when no prior information is available about the specific interacting regions within a disordered segment. Next, we used EOM to capture the intrinsic flexibility of frHMGB1 within the complex and to describe the size distribution of possible multiple conformations of frHMGB1•CXCL12. Herewith, we identified two EOM conformational ensembles fitting well the experimental data (with $x^2 < 1$, Supplementary Table 4). In the first CXCL12 binds to BoxA of frHMGB1, adopting alternatively open or more collapsed conformations (Fig. 7c). In the second one, the acidic

**Table 1 | Thermodynamic parameters of the interactions between CXCL12 and Ac-pep, frHMGB1 and frHMGB1-TL**

| | ITC[a] | | | | MST | |
|---|---|---|---|---|---|---|
| | $Kd_1$ (µM) | $Kd_2$ (µM) | $\Delta H_1$ (kcal/mol) | $\Delta H_2$ (kcal/mol) | $Kd_{[NaCl]=20mM}$ (µM) | $Kd_{[NaCl]=150mM}$ (µM) |
| Ac-pep | 0.6 ± 0.1 | 0.1 ± 0.1 | −10.4 ± 0.8 | 1.6 ± 0.7 | 0.6 ± 0.1[b] | 5.0 ± 0.6[b] |
| frHMGB1 | 1.2 ± 0.4 | $7.8 \times 10^{-3} \pm 1.5 \times 10^{-2}$ | −1.4 ± 0.1 | −2.1 ± 0.2 | 1.7 ± 0.2[c] | 31.8 ± 1.4[c] |
| frHMGB1-TL | n.d. | n.d. | n.d. | n.d. | 12.5 ± 5.5[c] | 81.4 ± 6.8[c] |

[a]CXCL12 is the titrant, $n = 2$ independent replicates, error estimates from covariance matrix.
[b]CXCL12 is the titrant, $n = 3$ independent replicates, values are mean ± standard deviation.
[c]frHMGB1 or frHMGB1-TL is the titrant, $n = 3$ independent replicates, values are mean ± standard deviation.
*n.d.* not detected.

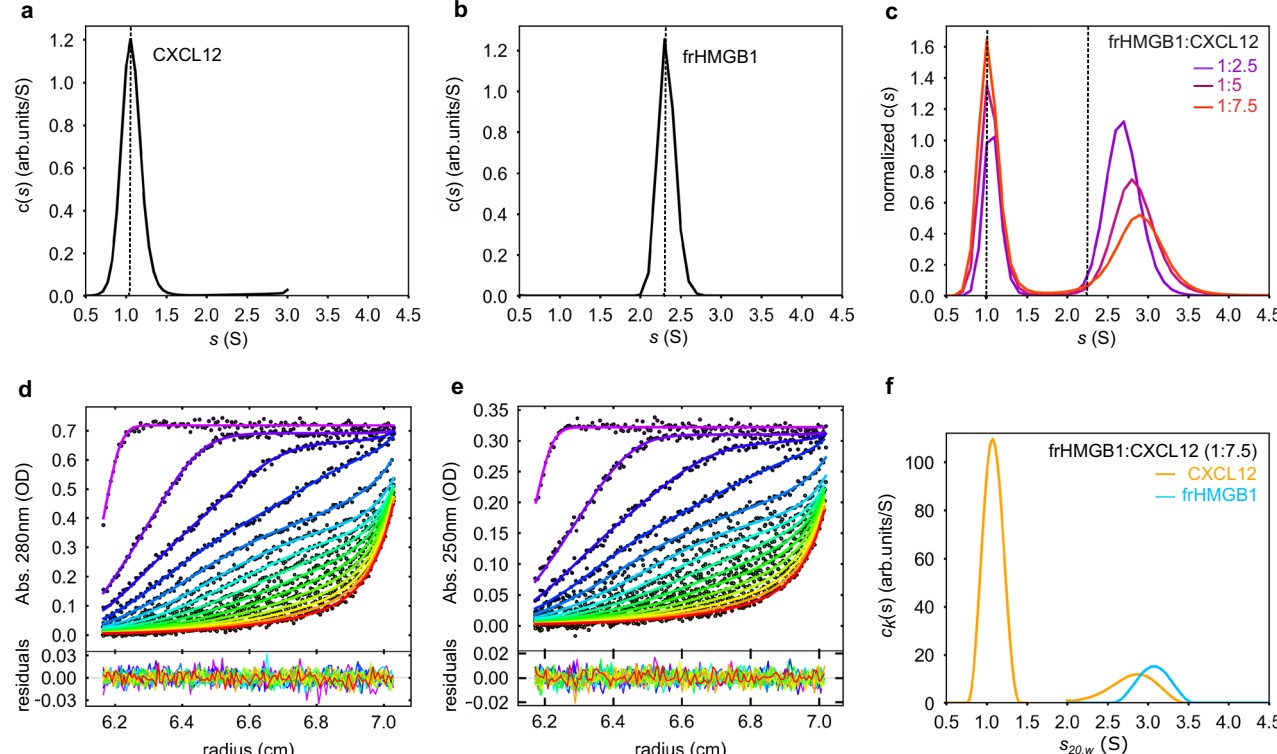

**Fig. 6 | frHMGB1 and CXCL12 form a transient 1:1 heterocomplex.** Sedimentation coefficient distributions c(*s*) of (**a**) free CXCL12 (38.2 µM) and (**b**) frHMGB1 (15.6 µM), scanned by absorbance at 280 nm. **c** Overlay of normalized c(*s*) showing the interaction between frHMGB1 (7.8 µM) and increasing concentrations of CXCL12 (colors). The dotted lines indicate the sedimentation coefficients of the free components. **d**, **e** Global multi-signal sedimentation velocity analysis to determine the stoichiometry of frHMGB1:CXCL12 complex, with 7.7 µM frHMGB1 and 43 µM CXCL12. The raw sedimentation signals of frHMGB1:CXCL12 mixture acquired at different time points with (**d**) absorbance at 280 nm, and (**e**) absorbance at 250 nm with the corresponding signal profiles as a function of radius in centimeters. The time-points of the boundaries are indicated in rainbow colors, progressing from purple (early scans) to red (late scans). Only every 3rd scan used in the analysis are shown. Residuals of the fit are shown at the bottom. **f** Decomposition into the component (*k*) sedimentation coefficient distributions, $c_k(s)$, for CXCL12 (yellow line) and frHMGB1 (cyan line). Source data are provided as a Source Data file.

IDR of frHMGB1 interacts with CXCL12, while the two HMG boxes adopt different reciprocal orientations (Fig. 7d).

Overall, SAXS analysis supports the notion that frHMGB1 and CXCL12 form an equimolar fuzzy complex.

**The acidic IDR modulates frHMGB1•CXCL12 binding to CXCR4**
Having shown that the acidic IDR of frHMGB1 plays a major role in the interaction with CXCL12, we tested whether it also played a role in the binding of the heterocomplex to its CXCR4 receptor. For these experiments, we chose mouse AB1 cells, a cellular model of malignant mesothelioma[49], as they express high levels of CXCR4[23].

We first verified the existence of the frHMGB1•CXCL12 heterocomplex in association with CXCR4 on the cell membrane. Proximity ligation assays (PLA) between HMGB1 and CXCL12 clearly identified the frHMGB1•CXCL12 heterocomplex on the surface of AB1 cells, and the PLA signal was competed by increasing concentrations of AMD3100, a specific CXCR4 antagonist (Fig. 8a). To confirm that the heterocomplex binds CXCR4 we deleted both *Cxcr4* alleles in AB1 cells by CRISPR/Cas9 mutagenesis. As expected, PLA on the surface of AB1 *Cxcr4* KO cells detected only background levels of the HMGB1•CXCL12 heterocomplex (Fig. 8b). Then, we treated wildtype AB1 cells with increasing amounts of Ac-pep, which significantly decreased the amount of detectable heterocomplex, in line with the NMR observation that Ac-pep can bind CXCL12 and disrupts the frHMGB1•CXCL12 heterocomplex.

We then tested the binding to cell surface CXCR4 of heterocomplexes made with full-length or tailless frHMGB1 (frHMGB1-TL). Cells were washed at acidic pH to remove ligands bound to receptors, and then exposed to preformed mixtures of CXCL12 and full-length or frHMGB1-TL; cells were kept at 4 °C to prevent receptor

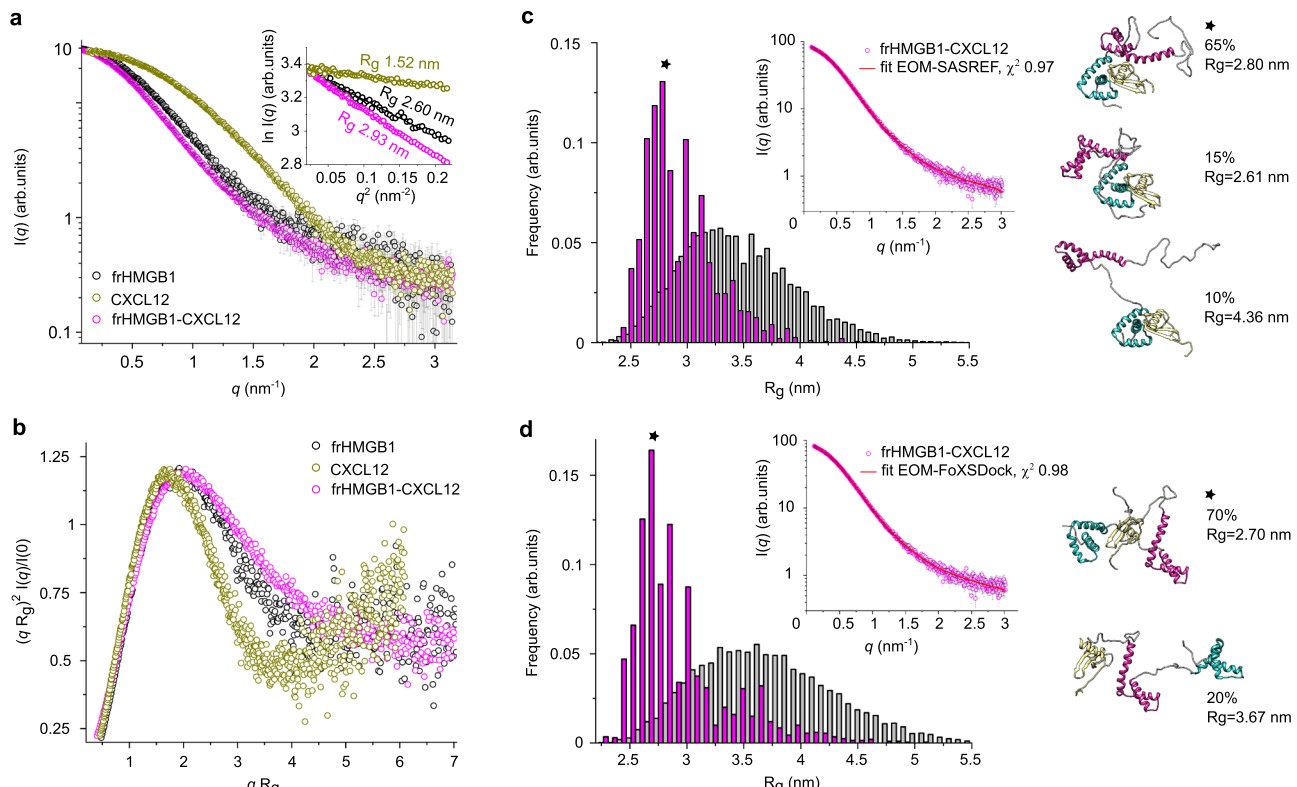

**Fig. 7 | SAXS studies of free frHMGB1, CXCL12 and of frHMGB1•CXCL12. a** $I(q)$ versus $q$ experimental SAXS profiles for CXCL12 (gold), frHMGB1 (black) and frHMGB1•CXCL12 complex (magenta). The curves are shifted by an arbitrary offset for better comparison. Error bars represent an estimate of the experimental error σ on the intensity recorded for each value of $q$ as assigned by data reduction software. In the inset, the Guinier regions used to estimate the radii of gyration ($R_g$, nm). **b** Dimensionless Kratky plots for the data presented in (**a**). Analysis of SAXS data by EOM on frHMGB1•CXCL12 models obtained using (**c**) SASREF and (**d**) FoXSDock with distributions of the selected ensemble conformers (magenta bars) and the initial pools of structures (grey bars) as a function of $R_g$ in nm. In the insets, $I(q)$ versus $q$ (magenta squares) with the EOM fitting (red lines with the corresponding $\chi^2$ values) for the frHMGB1•CXCL12 complex. Representative structures of the most populated EOM ensembles are shown in cartoon, with BoxA, BoxB, acidic IDR and CXCL12 coloured in cyan, magenta, grey and gold, respectively. For each ensemble, the frequency-weighted size average (the asterisks indicate the most populated fractions) and $R_g$ values are indicated. Source data are provided as a Source Data file.

internalization. Both full-length and tailless frHMGB1 formed complexes that could be detected by PLA on the cell surface; however, frHMGB1-TL formed fewer complexes and did not reach saturation (Fig. 8c).

Together, these results show that the frHMGB1 acidic IDR facilitates the binding of the frHMGB1•CXCL12 heterocomplex to its CXCR4 receptor, and that the Ac-pep competes with the acidic IDR for heterocomplex formation and binding to CXCR4.

## Discussion

Until now, structural studies aimed at getting mechanistic insights into frHMGB1•CXCL12 have focused on the interaction of CXCL12 with the single isolated HMG boxes[15,30]. In these studies the authors took advantage of NMR CSPs obtained from titrations of CXCL12 with the isolated HMG boxes to produce a 3D model in which each single HMG box bound one CXCL12 molecule[15,30]. The question arose whether such a simplified approach, in which the HMG boxes are separated and the IDRs (i.e. the acidic C-terminal tail and the linker connecting the HMG boxes) are neglected, can faithfully recapitulate the interaction of CXCL12 with the full-length protein. IDRs within their host proteins are often the major players in the recognition of their partners[50–52], and they are generally characterized by low complexity sequences, often containing charged residue[53]. In several cases IDRs maintain a significant level of disorder in their bound states, exchanging between multiple conformations and giving rise to so-called fuzzy complexes[31,32,37,54]. Their structural plasticity and ability to be engaged in fuzzy interactions renders IDRs well-suited to favour the formation

of biomolecular condensates through liquid-liquid phase separation, where specific interactions can be established without a clearly defined bound-state conformation[32]. Moreover, their conformational malleability allows IDRs to engage with various partners and in different cellular conditions, thus broadening the interaction scopes of their host protein[55]. This also applies for HMGB1, where the acidic C-terminal IDR modulates interactions with nucleic acids[56] and different proteins, such as histones[57,58] and p53[59]. Importantly, replacement of the acidic IDR with an arginine-rich basic tail has been recently shown to cause a complex human malformation syndrome, which results from HMGB1 aberrant phase separation in the nucleolus and nucleolar dysfunction[40].

In accordance with its functional relevance, here we show that the acidic IDR, whether isolated or in the context of the full-length protein, binds to the CXCL12 homodimerization surface. Interestingly, the CXCL12 residues involved in the interaction with both Ac-pep and frHMGB1 in part coincide with the ones involved in the interaction with negatively charged heparin oligosaccharides[60]. Importantly, the acidic IDR, because of its repetitive sequence, contains multivalent sites able to interact with CXCL12, as indicated by NMR titrations of $^{15}$N CXCL12 with synthetic peptide fragments of the acidic IDR. In line with this multivalency, NMR relaxation experiments performed on the $^{15}$N Ac-pep$_{rec}$ show that the acidic IDR maintains its conformational flexibility also when bound to CXCL12. Noteworthy, binding of full-length frHMGB1 to CXCL12 is significantly weaker than to the isolated acidic IDR, indicating competition of the intra-molecular frHMGB1 boxes with CXCL12 for the acidic IDR. Both ITC and MST experiments suggest

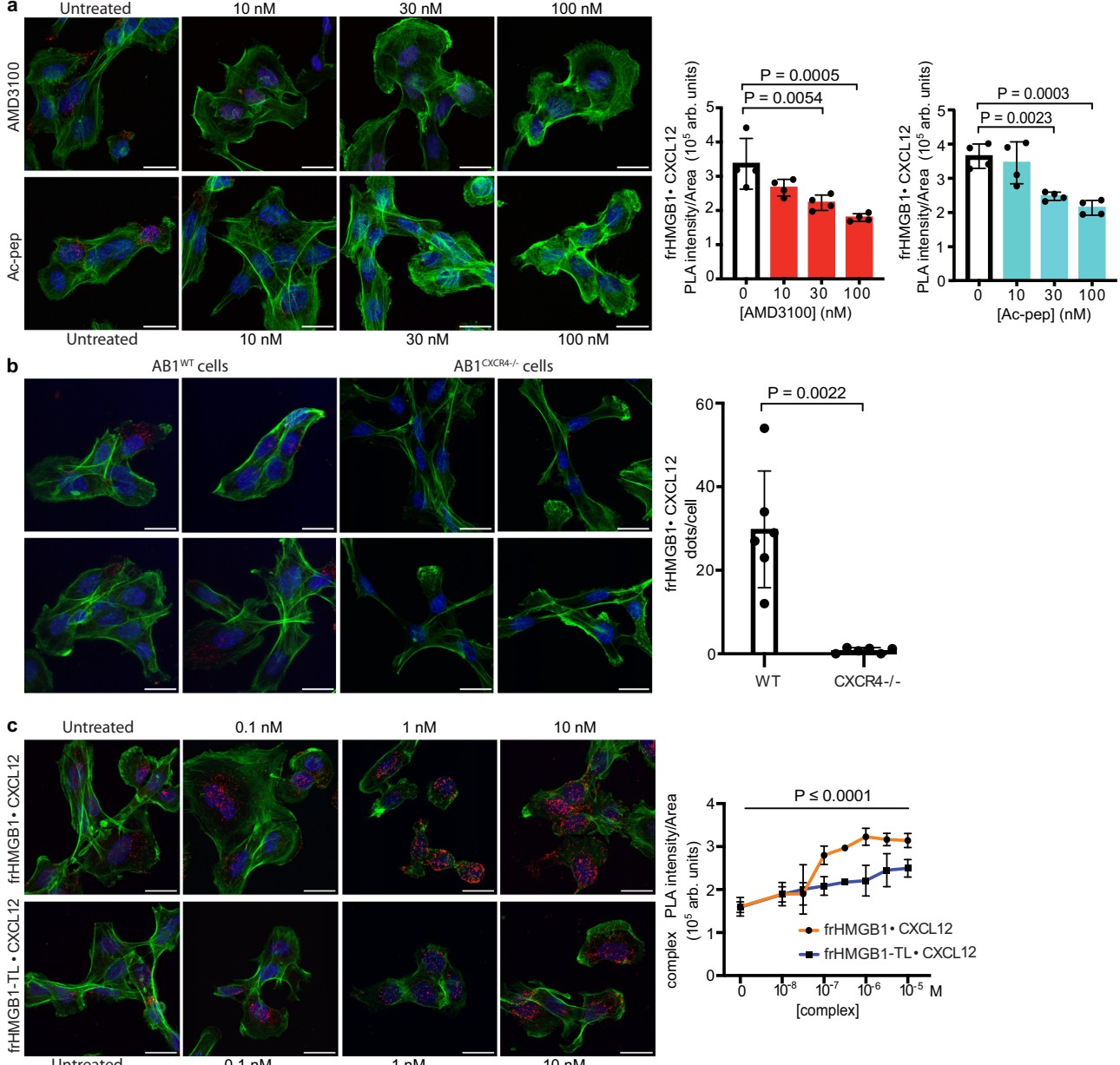

**Fig. 8 | The acidic IDR modulates frHMGB1•CXCL12 binding to CXCR4 on AB1 cells. a** Representative confocal microscopy images of Proximity Ligation Assays (PLAs) performed on the frHMGB1•CXCL12 complex on the surface of AB1 malignant mesothelioma cells. Cells were either untreated or treated with AMD3100 or Ac-pep (light blue bars) for 1 h at 37 °C/5% $CO_2$ at three different concentrations (10, 30, or 100 nM). PLA signal was quantified as described in the Methods section. Mean ± SD are indicated; $n = 4$ FOV (field of view) per concentration. One-way ANOVA was performed comparing AMD3100 (red) and Ac-pep (light blue) treated cells to untreated cells (white); AMD3100 versus untreated: **$P = 0.0054$; ***$P = 0.0005$; Ac-pep versus untreated: **$P = 0.0023$, ***$P = 0.0003$. Data are presented as arbitrary units (arb. units). Scale bar; 20 µm. **b** Representative confocal microscopy images of PLAs performed on the frHMGB1•CXCL12 complex on the surface of either wild type (WT) or *Cxcr4* knockout (*Cxcr4*[−/−]) AB1 malignant mesothelioma cells. PLA signal was quantified as number of dots per cell in FOV.

Statistical analysis; two-tailed, non-parametric Mann–Whitney test. Mean ± SD are indicated; $n = 6$ FOV per condition. Data are presented as arbitrary units (arb. units). Scale bar; 20 µm. **c** PLA signal quantification on the surface of ligand-stripped AB1 cells exposed at 4 °C to increasing concentrations of either frHMGB1•CXCL12 or frHMGB1-TL•CXCL12 equimolar heterocomplexes. Data are presented as arbitrary units (arb. units). Mean ± SD are indicated; $n = 4$ FOV per concentration. The difference between frHMGB1•CXCL12 (orange) and frHMGB1-TL•CXCL12 (blue) heterocomplexes is statistically significant ($P < 0.0001$) by two-way ANOVA. Sidak's multiple comparisons test revealed statistically significant differences between frHMGB1•CXCL12 and frHMGB1-TL•CXCL12 at the following concentrations; $10^{-7}$ M: $P = 0.0004$, $10^{-6.5}$ M, $P = 0.0003$, $10^{-6}$ M: $P < 0.0001$, $10^{-5.5}$ M: $P = 0.0013$, and $10^{-5}$ M: $P = 0.0032$. In all panels, nuclei are in blue (Hoechst 33342), phalloidin is in green, and the frHMGB1•CXCL12 PLA signal is red. Scale bar; 20 µm. Source data are provided as a Source Data file.

long range electrostatic interactions between the acidic IDR and the basic surface of CXCL12 as major drivers for binding, as increased ionic strength or deletion of the acidic IDR reduced binding to CXCL12. Long-range electrostatic interactions, though fundamental, are not the unique driving force for complex formation, as both NMR and MST indicate that tailless frHMGB1 (frHMGB1-TL) binds with micromolar affinity to the dimerization surface of CXCL12, albeit less well with

respect to the full-length protein. Thus, structured and unstructured frHMGB1 regions potentially recognize the same CXCL12 surface, which behaves as a structural hub for multivalent interactions. Notably, while previous NMR titrations suggested that the isolated HMG-boxes interact similarly with the dimerization surface of CXCL12[15], a preference for BoxA is apparent in the context of the tandem HMG-boxes, with the interaction surface mainly involving the two short

helices of the HMG domain. Of note, the targeting of BoxA is in line with the ability of CXCL12 to preferentially recognize the reduced forms of Cys-22 and Cys-44[20]. Moreover, our MST and NMR experiments show that, while dsHMGB1 is still able to bind to CXCL12, most likely via the acidic IDR, its affinity is starkly lower, thus confirming the important contribution of BoxA to the interaction. Privileged binding to BoxA is also in agreement with the equimolar stoichiometry of the heterocomplex suggested by both AUC and SAXS experiments.

Collectively, our data reveal that the frHMGB1•CXCL12 heterocomplex behaves as a typical fuzzy complex[31], whose formation relies on an intricate network of inter- and intra-molecular interactions of comparable affinities. As commonly observed in fuzzy binding, at least one of the elements, in this case the acidic IDR, is dynamic, maintains its conformational flexibility in the bound state and is fundamental for the interaction[36,61]. In addition, the intrinsic independent rotation of the two HMG-boxes provides an additional dynamic level to the system. Thus, the heterocomplex cannot be described by a unique structure, but is best represented by a heterogeneous ensemble of structures reflecting the different ongoing equilibria. Accordingly, the SAXS data of the heterocomplex are best fit by different plausible docking models obtained by EOM, where CXCL12 binds frHMGB1 in a promiscuous manner, alternatively associating to the manifold conformations of the acidic IDR and the different BoxA orientations.

Based on our data we propose a model (Fig. 9) in which the acidic IDR works as a wrapping antenna that recruits CXCL12 through long-range electrostatic interactions. Being intrinsically disordered, the acidic IDR does not present a single binding site to CXCL12 but rather resembles a diffuse binding cloud, in which multiple nearly-identical binding sites are dynamically distributed[62,63], preserving a significant flexibility even in bound states. This behavior is reminiscent of the mean electrostatic field created by multiple phosphates between the disordered yeast cyclin-dependent kinase (CDK) inhibitor Sic1 and its cognate binding partner, the F-box protein Cdc4[36]. The rapid on-off rate usually associated to long-range electrostatic interactions allows then CXCL12 to interact with HMG-boxes, in particular with BoxA in its reduced form, partially outcompeting HMGB1's intramolecular contacts. Remarkably, the binding mode of frHMGB1 to CXCL12 radically differs from the usual beta-beta or alpha-beta interactions observed in chemokine heterocomplexes[4].

CXCL12 bound to frHMGB1 can then accommodate inside the cradle formed by the transmembrane helices of CXCR4, yielding a three-component complex (Fig. 9), whose existence on the cell surface is supported by PLA experiments. These results are in line with the observation that the HMGB1•CXCL12 heterocomplex acts differentially from CXCL12 alone on the reorganization of the actin cytoskeleton and on β−arrestin recruitment[22]. Both actions depend on CXCR4. So far, however, our data do not specify whether a specific subset of the conformations of the HMGB1•CXCL12 heterocomplex preferentially binds to CXCR4. In our cellular context, the acidic IDR facilitates the binding of the frHMGB1•CXCL12 heterocomplex to CXCR4, and Ac-Pep does compete with it, confirming the results obtained by NMR, and implying that the acidic IDR plays a major role in the formation of the CXCR4•frHMGB1•CXCL12 ternary complex as well. The fuzziness of the interactions in the frHMGB1•CXCL12 heterocomplex forces us to reconsider the structure of HMGB1, which itself can be considered fuzzy: in solution HMGB1 populates an ensemble of different microstates in which the D/E repeats are associated through electrostatic transient interactions with different segments of HMGB1, that in turn are partially screened and only transiently exposed to natural interactors[26]. Plausibly, the fuzzy conformation of HMGB1 allows interactions with multiple partners[64], all of which have micromolar-range apparent affinities. Indeed, HMGB1 often works as a chaperone, by binding one interactor and facilitating its further interaction with another molecule[65–67]. As such, the mechanism and the conformational

heterogeneity through which HMGB1 binds to CXCL12 is in part reminiscent of other chaperones, like small heat shock proteins, that exploit their large IDR to transiently interact with their clients[68].

In brief, we show here that frHMGB1 and CXCL12 form a bimolecular heterocomplex, which is fuzzy and highly dynamic, and can go on to form a ternary complex with the CXCR4 receptor. These conclusions are in line with the concepts of HMGB1 working as a molecular chaperone and of IDPs/IDRs being typically involved in signaling, regulation, recognition, and control of various cellular pathways[69].

Finally, recent studies have revealed that inhibiting heterophilic interactions among chemokines interactions can reduce inflammation[13]. Consequently, targeting these protein-protein interactions is emerging as a valuable strategy for developing selective antagonists to finely modulate specific inflammatory responses. This holds true also for the frHMGB1•CXCL12 heterocomplex, a key regulator of inflammatory cell recruitment via the CXCR4 axis, making it an attractive target for selective anti-inflammatory agents. Indeed previous research has demonstrated that HMGB1•CXCL12 is druggable exploiting specific pockets within HMGB1 and CXCL12[24]. As the targeting of IDR and IDPs is starting to emerge as a pharmacological strategy[70], we anticipate that interfering with the fuzzy interactions within the HMGB1•CXCL12 heterocomplex could represent an additional opportunity to inhibit its detrimental activity in inflammatory conditions.

## Methods
### Protein production and synthetic peptide
Recombinant labeled and unlabelled (15N) HMGB1 (uniprot code P63159, residues 2–215, renumbered from 1 to 214) and HMGB1-TL (uniprot code P63159, residues 2–188, renumbered from 1 to 187) were transformed in BL21 (DE3) pLysS and BL21 (DE3) strains of *E. coli*, respectively, using the pETM-11 expression vector (EMBL, Heidelberg, DE). The proteins were purified as described in Supplementary Information. dsHMGB1 and dsHMGB1-TL were obtained by extensively dialyzing their fully reduced forms against a buffer devoid of Dithiothreitol (DTT, volume ratio sample:buffer 1:2000), changing the buffer every 2 h overday, followed by an overnight dialysis. To reach the fully oxidized state, the samples were incubated at 4 °C for 72 h, before NMR and MST analysis. The oxidation status of HMGB1 constructs was checked by 1H-1D NMR monitoring the aliphatic spectral region, which displays distinctive features for the oxidized and reduces forms. Recombinant labeled and unlabelled 15N/13C CXCL12 (uniprot code P48061, residues 23–89, renumbered from 1–68) and CXCL12-LM were transformed into *E. coli strain* BL21 (DE3) using the expression vector pET30a. The proteins were purified as described in Supplementary Information. Unlabeled N-terminal-6His-tagged CXCL12 for MST measurements was provided by HMGBiotech (Milan, Italy).

For the production of CXCL12-LM, site-directed mutagenesis was performed to introduce mutations L55C and I58C in CXCL12 pET30a expression vector by using standard overlap extension methods (Supplementary Information). The DNA constructs were sequenced by Eurofins (Milan, Italy). Protein concentrations were determined considering molar extinction coefficients at 280 nm of 21430, 8730 $M^{-1}$ $cm^{-1}$, 8855 $M^{-1}$ $cm^{-1}$ for HMGB1 (and HMGB1-TL), CXCL12 and CXCL12-LM, respectively.

Recombinant Ac-pep (Ac-pep$_{rec}$) (corresponding to the HMGB1 acidic IDR, 30 aa, uniprot code P63159, residues 186–215, renumbered from 185 to 214) was produced as 6His-SUMO3 tagged fusion protein using the pETM11-SUMO3 expression vector (EMBL, Heidelberg, DE). Cloning was performed by GENEWITZ from Azenta Life Sciences. Expression was carried out in BL21 (DE3) *E. coli* cells by induction at 30 °C for 18 h with 0.5 mM Isopropil-β-D-1-thiogalattopiranoside (IPTG). Cells were harvested by centrifugation (8200 × g, 4 °C) and lysed by sonication in Lysis Buffer (150 mM NaCl, 20 mM Tris-HCl pH 8, 0.2% NP-40, 2 mM β-mercaptoethanol (BME), Complete EDTA-free

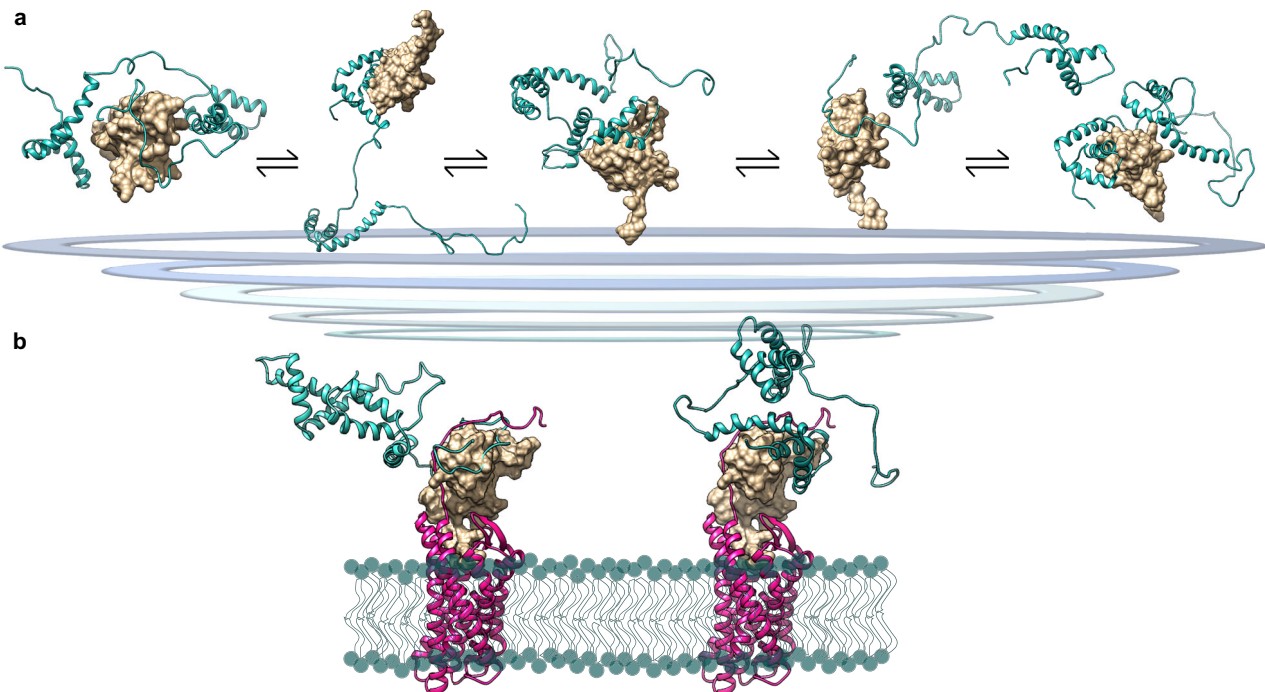

**Fig. 9 | Model of frHMGB1•CXCL12 fuzzy complex. a** Representative EOM models of the fuzzy interactions between frHMGB1 (cyan cartoon) and CXCL12 (gold surface). **b** Explicative representations of possible different frHMGB1•CXCL12 conformations bound to CXCR4. Two SAXS-EOM frHMGB1•CXCL12 models have been superimposed on the theoretical model of CXCL12 in complex with CXCR4[85], with CXCR4 in magenta, CXCL12 in gold surface and frHMGB1 in cyan; the lipid bilayer is represented with cyan spheres and lines.

protease inhibitor (Roche), 2 μg/mL DNAse, 20 μg/mL RNAse). The 6His-SUMO3-tagged protein was purified on a histidine affinity column (Ni-NTA agarose, Quiagen) and eluted with Elution Buffer (150 mM NaCl, 20 mM Tris-HCl pH 8, 2 mM BME, 300 mM imidazole). The 6His-SUMO3 tag was cleaved during overnight dialysis at 4 °C against Dialysis Buffer (20 mM Tris-HCl pH 8, 150 mM NaCl, 2 mM BME, 5% glycerol) by addition of 6His-tagged Small Ubiquitin-Related Modifier (SUMO)-Specific Protease 2 protease (SENP2, homemade). 6His-SUMO3 tag and 6His-SENP2 were then separated from the digested Ac-pep$_{rec}$ by a second purification step on a Ni-NTA column. The peptide (having a serine at the N-terminus, deriving from cloning) was further purified on an HiTrap Q HP anion exchange chromatography column (Cytiva) using buffer A (20 mM Tris-HCl pH 8, 50 mM NaCl) and a slow linear gradient (3 ml/min flow, 15 column volumes) of buffer B (20 mM Tris-HCl pH 8, 1 M NaCl). The purified peptide was dialyzed against Milli-Q, lyophilized and resuspended into NMR buffer (20 mM NaH$_2$PO$_4$/Na$_2$HPO$_4$ pH 6, 20 or 150 mM NaCl). The peptide identity was confirmed by mass spectroscopy. Peptide concentration was determined using the absorbance at 205 nm and the molar extinction coefficient at 205 nm (83,400 M$^{-1}$ cm$^{-1}$) calculated by the web server [http://nickanthis.com/tools/a205.html]. Uniformly labeled $^{15}$N and $^{15}$N/$^{13}$C Ac-pep$_{rec}$ was produced by growing BL21 (DE3) *E. coli* cells in M9 minimal medium containing $^{15}$NH$_4$Cl, with or without $^{13}$C D-glucose.

Synthetic Ac-pep (corresponding to HMGB1 acidic IDR, 30 aa, uniprot code P63159, residues 186–215, renumbered from 185 to 214) was purchased from Caslo Lyngby, Denmark. Synthetic Ac-pep fragments corresponding to residues 185–195 (Ac-pep$_{185-195}$) and residues 204–214 (Ac-pep$_{204-214}$) and the synthetic peptide corresponding to the arginine rich tail present in an aberrant HMGB1 form[40] (R$_{185}$RKMRKMKRMRRRKMKKMKMKKKMKKMNK$_{214}$, whereby M$_{210}$-M$_{211}$ where mutated into lysines to avoid possible aggregation problems deriving from four consecutive methionines) were purchased from Davids Biotechnologie GmbH (Germany). Peptide purity

(>95%) was confirmed by HPLC and mass spectrometry. Peptide concentration was estimated from its dry weight. For ITC measurements with Ac-pep, to obtain a more accurate estimation of the concentration by UV a tyrosine ($\varepsilon_{274nm}$ = 1405 M$^{-1}$ cm$^{-1}$) was added at the peptide *N*-terminus, for MST experiments 5,6 FAM (5(6)-Carboxyfluorescein) was added at the peptide N-terminus.

## NMR spectroscopy (NMR)
NMR experiments were performed at 298 K on a Bruker Avance 600 MHz equipped with inverse triple-resonance cryoprobe and pulsed field gradients (Bruker, Karlsruhe, Germany). Typical samples concentration was 0.1–0.4 mM. Data were processed using NMRPipe 10.9[71] or Topspin 3.26 (Bruker) and analyzed with CCPNmr Analysis 2.4[72]. The $^1$H, $^{13}$C, $^{15}$N chemical shifts of CXCL12 in the presence of Ac-pep and of CXCL12-LM were obtained from three-dimensional HNCA, CBCA(CO)NH, CBCANH, HNCO experiments (Supplementary Fig. 14, 96% amides assigned).

## NMR titration experiments
Before NMR titrations the samples (titrant and titrated solution) were dialyzed against the same buffer, 20 mM NaH$_2$PO$_4$/Na$_2$HPO$_4$ pH 6.3, 20 mM NaCl, supplied with 0.15 mM 4,4-dimethyl-4-silapentane-1-sulfonic acid (DSS) and D$_2$O (10% v/v). Of note, HMGB1 upon removal of DTT maintained its thiol form for at least 12 h. In the case of titrations with synthetic peptides the lyophilized peptides were dissolved directly in the NMR buffer, where necessary the pH was adjusted with 0.1 M NaOH. Titrations were carried out by adding to $^{15}$N labelled protein samples (typically 0.1 mM) small aliquots of concentrated (15 mM) peptide stock solutions or unlabelled protein (0.6–1 mM). For each titration point (0.5, 1, 1.5, 2 equivalents of ligand) a 2D water-flip-back $^{15}$N-edited HSQC spectrum was acquired with 2048 (160) complex points, apodized by 90° shifted squared (sine) window functions and zero filled to 2048 (512) points for $^1$H ($^{15}$N). Spectra assignment was made following individual cross-peaks through the titration series. For

each residue the weighted average of the $^1$H and $^{15}$N chemical shift perturbation (CSP) was calculated as

$$CSP = \sqrt{\frac{\Delta\delta H^2 + \Delta\delta N^2/25}{2}} \qquad (1)$$

where $\Delta\delta H$ and $\Delta\delta N$ are, respectively, the differences of $^1$H and $^{15}$N chemical shifts between free and bound protein[73]. The corrected standard deviation ($\sigma_O$) was calculated as described in[74]. Because of extensive line broadening due to ligand binding in the intermediate exchange regime on the NMR time scale, we also monitored changes in the intensity ratio ($I/I_O$) of the $^1$H-$^{15}$N amide resonances, where $I_O$ and $I$ are the peak intensities in the free and bound protein, respectively.

## Lineshape analysis
We performed 2D NMR lineshape analysis using the software TITAN 1.6[33]. Spectra were processed with NMRpipe 10.9[71] with a script provided by TITAN 1.6. A series of regions of interest containing isolated peaks with CSPs > avg + SD were selected (V18, V23, K24, H17, A40, H25 and R12 for $^{15}$N-CXCL12 and V23, K24, H25 for $^{15}$N-CXCL12_LM) (Supplementary Fig. 2b) and fitted by optimizing the chemical shifts and line widths for the free and the bound state. The program estimates $K_d$, $k_{off}$ (we fixed 1:1 stoichiometry). Error estimates for the fit parameters were obtained using the bootstrap resampling of residuals procedure implemented in TITAN 1.6[33].

## NMR relaxation measurements
Measurements of heteronuclear {$^1$H}-$^{15}$N nuclear Overhauser (NOE) enhancement, longitudinal and transversal $^{15}$N relaxation rates ($R_1$, $R_2$) on free $^{15}$N Ac-pep (0.3 mM) and in the presence of CXCL12 (1:1) were performed using standard $^1$H–$^{15}$N HSQC spectra with varying relaxation delays[75]. Duty-cycle heating compensation were used for both $T_1$ and $T_2$ relaxation experiments[76]. $T_1$ and $T_2$ decay curves were sampled at 10 and 9 different relaxation delays, respectively and two duplicate delays (ms). ($T_1$: 50/50, 150, 250/250, 450, 650, 900, 1100, 1400, 2000, 3000 and $T_2$: 72, 12/12, 28, 44, 56, 112/112, 144, 200, 244) collected in random order, with 4 s recovery delay. Two delays were acquired twice for evaluation of the average ⟨$T_1$⟩, ⟨$T_2$⟩ values and the corresponding standard deviations. The {$^1$H}-$^{15}$N NOEs were measured recording HSQC spectra with and without proton saturation in an interleaved fashion using a 6 s recycle delay/saturation. The standard deviation of the noise of both saturated and unsaturated spectra were used to estimate $I_{sat}/I_{unsat}$ uncertainty via error propagation formula. $R_1$, $R_2$ and {$^1$H}-$^{15}$N NOEs values have been obtained using the CcpNmr Analysis 2.4 fitting routine[72] and internal scripts. Relaxation experiments have been repeated twice on $n = 2$ biologically independent samples.

## Isothermal titration calorimetry (ITC)
Proteins and peptides were dialyzed in a Slide-A-lyser mini-dialysis unit with a 2000 MWCO and Biodialyzer with 500 Da MWCO (Harvard Apparatus, US) against 20 mM TrisHCl at pH 7.5, 50 mM NaCl (or 150 mM NaCl when explicitly stated). ITC data were collected on a MicroCal PEAQ-ITC instrument (Malvern). The cell temperature was set to 37 °C, the syringe stirring speed to 750 rpm, and reference power to 10 µcal/sec. frHMGB1 (frHMGB1-TL, Ac-pep) and CXCL12 were loaded into the cell and syringe at concentrations of ~10 and ~600 µM, respectively. The MicroCal PeakITC software (Malvern) was applied for initial data analysis. For global fitting, thermograms were integrated using NITPIC 2.0.0[38] and SEDPHAT 15.2b[77]. Data were fit with the two non-symmetric sites microscopic $K$ model[39] applying Simplex and Marquardt-Levenberg optimization algorithms, as implemented in SEDPHAT 15.2b. Error estimates were based on the covariance matrix generated by the Marquardt-Levenberg algorithm. Experiments have been repeated on $n = 2$ independent samples.

## Microscale thermophoresis (MST)
MST experiments were performed at 24 °C on a NanoTemper® Monolith NT.115 instrument. Binding between Ac-pep and CXCL12 was monitored titrating CXCL12 (16-points) into 50 nM N-terminal-5,6-FAM-labelled Ac-pep (CASLO ApS, Denmark), using the blue filter, 20% LED power and medium MST power. Binding between frHMGB1, dsHMGB1 or frHMGB1-TL and CXCL12 was monitored titrating the different HMGB1 proteins (16-points) into 50 nM 6His-tagged CXCL12, non-covalently labelled with the NT-647 conjugated tris-NTA (RED-tris-NTA) fluorescence dye, using the red filter, 40% LED power and medium MST power. Before MST titrations the proteins (the ligand and the fluorescently labelled target) were dialyzed against the same buffer, 20 mM NaH$_2$PO$_4$/Na$_2$HPO$_4$ pH 7.3, 0.05% TWEEN and 20 mM or 150 mM NaCl. In the case of 5,6-FAM-labelled Ac-pep, the lyophilized peptide was dissolved directly in the MST buffer and the pH adjusted to pH 7.3.

The 16 titration points of each experiment were made through serial dilution of the ligand stock into MST buffer and then addition of a constant amount of fluorescently labelled target (50 nM). Before mixing, both the ligand and the fluorescently labelled target were centrifuged at 15,000 g, 4 °C, for 10 min. Maximum concentrations of frHMGB1, frHMGB1-TL, dsHMGB1 and CXCL12 ligands in the titrations were 104–343 µM, 315 µM, 300 µM and 287–295 µM, respectively. Complex samples were incubated for 30 min before loading into Nano-Temper premium capillaries. Each experiment was repeated on $n = 3$ independent samples, data points are the average of the triplicates and the error bars correspond to the standard deviation.

Since CXCL12 addition induced >10% variation in the fluorescence of 5,6-FAM-Ac-Pep, thermophoresis traces could not be used to measure binding affinity, we therefore used the quenching of the 5,6-FAM-Ac-Pep fluorescence upon binding to estimate the $Kd$. Data analyses were carried out using NanoTemper Analysis 2.3 software and the $Kd$ model fitting (one binding site).

## Analytical ultracentrifugation (AUC)
Sedimentation velocity experiments were performed on an Optima XLI (Beckman Coulter) using an A50 Ti eight-hole rotor and with seven 400 µl samples in standard dual-sector Epon centerpieces equipped with sapphire windows. Absorbance data were acquired at 250 and 280 nm simultaneously with the absorbance scanner in the continuous mode with radial increments of 0.003 cm. Three assembled centrifugation cells containing, respectively, free CXCL12 (or CXCL12-LM) (38.2 µM or 37.6 µM), frHMGB1 (15.6 µM), frHMGB1-TL (15.6 µM) and the frHMGB1:CXCL12 (or frHMGB1:CXCL12-LM) mixture at the loading ratios of 1:2.5, 1:5, 1:7.5 (with frHMGB1 or frHMGB1-TL at 7.8 µM concentration), pH 7.5, 20 mM TrisHCl, 50 mM NaCl (or 150 mM NaCl when explicitly stated), were equilibrated at 20 °C under vacuum for approximately 1.5 h prior starting the experiment. Subsequently, centrifugation was performed at 163,000.44 g with 90 scans. The highest protein concentration was determined by the absorbance for which the linear relationship according to Lambert-Beer Law was still guaranteed (O.D. max = 1.0). The buffer density and viscosity were estimated using SEDNTERP 1.0[78].

The sedimentation coefficient distributions ($c(s)$) at 280 nm of the single proteins were obtained by applying the diffusion-deconvoluted $c(s)$ model, implemented in SEDFIT 16.1c[41]. Concentration profiles in terms of absorbance ($a(r,t)$) were modelled as the sum of Lamm Equation solutions scaled by a continuous distribution $c(s)$ as follows:

$$a(r, t) \cong \int_{s_{min}}^{s_{max}} c_k(s)\chi(s, D_k(s), r, t)ds \qquad (2)$$

where $s$ is the sedimentation coefficient, $\chi$ ($s$, D($s$), $r$; $t$) is the Lamm Equation solution that is dependent on D($s$), the corresponding diffusion coefficient, $r$, radius from the center of rotation, and $t$, the time from the beginning of the experiment.

Multisignal sedimentation velocity (MSSV) analysis[44] was performed to determine the stoichiometry of the complex formed by frHMGB1 and CXCL12 (or CXCL12-LM). It is important to note that in MSSV, like in c(s) model, the data are not fitted a priori to a binding stoichiometry model (e.g. 1:1 or 1:2) and no binding model assumption is made. The experimental sedimentation data, whose physical observable is absorbance, are fitted with mathematical equations (the Lamm equations) describing the sedimentation process without assuming any binding stoichiometry. The sedimentation process is described as a superposition of normalized Lamm equation solutions of ideally sedimenting species at a range of sedimentation coefficients $s(1... N)$, with N indicating the range of molecules size expressed as sedimentation coefficient in Svedberg and using the hydrodynamic scaling law to estimate the corresponding diffusion coefficients. The Lamm equation solutions once superimposed give the number of species, molecular weight and weight averaged shape. More in detail, in MSSV the standard c(s) approach is modified to deconvolute the contributions of individual species in a component distribution $c_k(s)$ where $k$ represents the individual components of a mixture. The absorbance at wavelength $\lambda$ ($a_\lambda(r,t)$) is modelled as:

$$a(r,t) \cong \sum_{k=1}^{K} \epsilon_\lambda^{k_l} \int_{s_{min}}^{s_{max}} c(s)\chi(s,D(s),r,t)ds \qquad (3)$$

where $l$ is the path length, $k$ is the number of solutes present, and $c_k(s)$ is a continuous distribution for component $k$.

MSSV deconvolution is possible when the complex components have sufficiently different spectral properties[44], i.e.

$$D_{norm} = \frac{||\det \epsilon_k^\lambda||}{\prod_k ||\vec{\epsilon}_k||} > 0.065 \qquad (4)$$

As the molar extinction coefficients of frHMGB1 ($\epsilon_{280} = 20872.8\,M^{-1}\,cm^{-1}$, $\epsilon_{250} = 7987.7\,M^{-1}\,cm^{-1}$) and CXCL12 ($\epsilon_{280} = 9907.2\,M^{-1}\,cm^{-1}$, $\epsilon_{250} = 4704.2\,M^{-1}\,cm^{-1}$) or CXCL12-LM ($\epsilon_{280} = 8413.4\,M^{-1}\,cm^{-1}$, $\epsilon_{250} = 4840.5\,M^{-1}\,cm^{-1}$) delivered sufficient spectral discrimination with $D_{norm} > 0.08$ (for frHMGB1:CXCL12) and $D_{norm} > 0.16$ (for frHMGB1:CXCL12-LM), it was possible to use SEDPHAT 15.2b to perform global multi-signal analysis of the sedimentation boundary associated to the co-sedimenting complex[44]. MSSV of the frHMGB1:CXCL12 (7.7 μM:43 μM) and frHMGB1:CXCL12-LM (6.9 μM:51 μM) heterocomplexes were collected at 250 nm and 280 nm and globally fitted using the multi-wavelength discrete/continuous distribution analysis with mass constraints in SEDPHAT 15.2b[44]. Integration of the resulting $c_k(s)$ distributions revealed the content of each protein component under the peak at a given sedimentation value. Plots of the signal profiles, fits, residuals and the MSSV results were generated using GUSSI 1.4.2[79].

**Small angle X-ray scattering (SAXS)**
The experiments were performed at the ESRF bioSAXS beamline BM29, Grenoble, France at a detector distance of 2.869 m. CXCL12 and frHMGB1 were measured in batch mode at 20 °C using the sample changer immediately after protein thawing and centrifugation (30 min at 16,000 g); 45 μL of sample solution at three different concentrations (1.95, 3.0 and 3.9 mg/mL per each protein, 20 mM Tris pH 7.5, 50 mM NaCl) were used. To characterize the frHMGB1•CXCL12 complex we tested the following protein molar ratio conditions: 1:1, 1:2, 1:4 and 1:6 (frHMGB1:CXCL12, with frHMGB1 1.95 mg/mL). After incubation and centrifugation, the supernatants were immediately measured in the SAXS beamline. $n = 10$ frames of 0.5 s each were collected for each sample (frHMGB1•CXCL12, free components at different protein concentration). Data from the frHMGB1 and CXCL12 dilutions were merged following standard procedures to create an idealized scattering curve, using PRIMUS within ATSAS 3.2.1[80]. The pair distribution function P(r) was calculated using GNOM version 5.0 (r14886)[81].

Protein molecular masses were estimated using both Porod volume and scattering mass contrast methods. All the plots were generated using OriginPro (Version 2022, OriginLab Corporation, Northampton, MA, USA).

To model the frHMGB1•CXCL12 heterocomplex, we employed a multistep approach relying on complementary experimental knowledge (NMR and AUC). The SAXS curve of frHMGB1 in the presence of two equivalents of CXCL12 was used, as in this condition frHMGB1 is almost fully saturated and the contribution of free CXCL12 to the scattering signal is negligible, as also confirmed by OLIGOMER analysis (within ATSAS 3.2.1[80]) (Supplementary Methods and Supplementary Fig. 15). This SAXS curve was used to generate with EOM a representative frHMGB1 starting structure for subsequent rigid docking on CXCL12 (pdb: 2KEE). In particular, two initial rigid docking models were generated. The first one was obtained with SASREF[47] (within ATSAS 3.2.1[80]) combining the solution scattering data and guiding the docking of CXCL12 (residues 23–28 and 66–67) on frHMGB1 (residues 15–45), as suggested by NMR experiments. The second model was obtained docking CXCL12 (residues 23–28 and 66–67) onto HMGB1 acidic IDR performing a global search with FoXSDock version main.ec6dbc2[48] (Supplementary Fig. 13). Next, to describe the dynamic and fuzzy nature of the heterocomplex, we fixed the protein-protein interaction surfaces obtained with SASREF[47] and FoXSDock version main.ec6dbc2[48], and allowed the rest of frHMGB1 to explore a wide range of conformations using the EOM algorithm within ATSAS 3.2.1[80]. Similarly, frHMGB1 alone was modelled using the EOM approach (parameters for the analysis are in Supplementary Table 4). The missing residues connecting the rigid bodies and the modelled segments were added with MODELLER 10.1[82]. The $\chi^2$ values of EOM fits over the 0.1–3 nm$^{-1}$ q range of experimental SAXS curves were determined with CORMAP within ATSAS 3.2.1[80].

All SAXS data were deposited into SASBDB data bank (CXCL12, SASDB ID: SASDRG9); frHMGB1, SASDB ID: SASDRH9; frHMGB1•CXCL12, SASDB ID: SASDRJ9.

**Molecular images**
Molecular images were generated by PyMOL Molecular Graphics System, open source version, Schrödinger, LLC and UCSF Chimera 1.16[83] and assembled with Inkscape 0.92.

**Cell line and treatments**
AB1 mouse malignant mesothelioma cells (MM; Cell Bank, Australia) were cultured in RPMI 1640 (Life Technologies, UK), supplemented with 5% v/v fetal bovine serum (FBS; Life Technologies, UK), 2 mM L-glutamine, and 100 U/ml penicillin/streptomycin at 37 °C; 5% CO₂. Cells were not cultured past passage 10 after cell thawing. To show binding of the HMGB1•CXCL12 heterocomplex to CXCR4, cells were incubated or not at 37 °C with increasing concentrations of AMD3100 and then fixed and quantified for the PLA signal. To validate these observations, we generated a *Cxcr4* knockout AB1 cell line (see further) to determine whether CXCR4 is indeed required for frHMGB1•CXCL12 complex formation. Untreated *Cxcr4−/−* and wild-type AB1 cells were fixed and quantified for the PLA signal. As a further verification, wild-type and *Cxcr4−/−* cells were co-cultured at a 3:1 and a 1:3 ratio, fixed, and quantified for the PLA signal to demonstrate that the ratio of cells with high and low PLA signal would be 3:1 (or 1:3), confirming the previous experiment (Supplementary Fig. 16).

To show the effect of Ac-pep on the binding of the frHMGB1•CXCL12 heterocomplex to the receptor, cells were incubated or not at 37 °C with increasing concentrations of Ac-pep and then fixed and quantified for the PLA signal.

To measure the binding of the frHMGB1•CXCL12 heterocomplex to its receptor, cells were washed with 10 mM Tris-HCl, pH 5.3, washed with RPMI and then incubated for 20 min at 4 °C with RPMI containing the indicated concentrations of preformed full-length or tailless

frHMGB1•CXCL12 heterocomplexes. Cells were then fixed and quantified for the PLA signal.

## Proximity ligation assay (PLA)

AB1 cells ($2 \times 10^4$) were seeded onto 15 mm glass coverslips and incubated overnight at 37 °C/5% $CO_2$, and treated the following day as described above. To perform PLA, cells were fixed for 10 min in 4% paraformaldehyde in PHEM buffer (1:1) at room temperature (RT), washed in 0.2% bovine serum albumin (BSA)/PBS and blocked with 10% goat serum in 4% BSA/PBS for 1 h at RT. Cells were then incubated overnight at 4 °C with two primary antibodies: mouse monoclonal anti-HMGB1 (1:1000; HMGBiotech, HM-901) and goat polyclonal anti-CXCL12 (1:50; R&D Systems, #AF-310-NA). Following three washes with 0.2% BSA/PBS, cells were incubated with secondary oligonucleotide-linked antibodies for 1 h at 37 °C (anti-mouse PLUS [#DUO92001] and anti-goat MINUS [#DUO92006], Duolink, Sigma-Aldrich; 1:5 dilution in Duolink antibody diluent [DUO82008]) and processed according to the Duolink PLA fluorescence protocol (Sigma). Finally, cells were stained with Phalloidin-FITC (1:500; Sigma-Aldrich, #P5282) and Hoechst (1 μg/mL; Sigma-Aldrich, #33342), and mounted on microscope slides with Flourosave reagent (Merck, #345789).

All fields of view (FOV) for PLA were acquired using a Leica TCS SP5 X confocal microscope (Leica Biosystems) with a 63x objective, using channels for Hoechst (405 nm), FITC (488 nm), and Texas Red (561 nm). Z-step size was set at 0.69 μm and the top and bottom of the cells were ascertained manually prior to acquiring each image. Each image was stacked to max intensity using ImageJ/Fiji software and saved as tiff files. These images were then used to create cytoplasmic masks using Cellpose[84]. Python 3.10.0 was used to run Cellpose. Cytoplasmic masks were used to ascertain cytoplasmic specific PLA signal in ImageJ/Fiji. Data are expressed as either as the average number of PLA dots per cell (Fig. 8b) or as Raw Integrated Density (sum of all pixel intensities in region of interest, in arbitrary units) normalized to the area of the respective cytosolic regions (Fig. 8a, c where the number of dots was difficult to count due to overlaps). All PLA experiments have been repeated twice. GraphPad Prism, version 8.4.0 was used for the generation of original graphs and statistical tests.

## Generation of *Cxcr4* knockout AB1 cells

*Cxcr4* knockout AB1 cells were generated using the LentiCRISPRv2 *BsmB*I CRISPR/Cas9 system to obtain two cuts flanking the last exon of mouse *Cxcr4* gene. Guide RNAs were designed using the UCSC Genome Browser and Primer3 program (Sg1FWD: TGTTTGGTTATGCTGTGTG; Sg1REV: ACAAACCAATACGACACACT; Sg2FWD: TGAAATGGACGTTT TCATCC; Sg2REV: ACTTTACCTGCAAAAGTAGG). The LentiCRISPRv2 vector was digested using *BsmB*I (#R0739; New England Biolabs), ligated with the oligonucleotide guides and transformed into Stbl3 bacterial cells, which were plated on ampicillin plates and screened for the correct CXCR4-sgRNA construct. Vector lentiviruses were then produced in HEK293 cells after transfection with lipofectamine 3000 of VSV-G envelope (#14888; Addgene), ps-PAX2 packaging (#12260; Addgene) and CXCR4-sg-RNA plasmids. AB1-B/c-LUC cells were then infected with this lentivirus and selected using Neomycin (10 μg/mL); single cells were plated in a 96-well plate and allowed to grow to confluence before DNA extraction and PCR with primers flanking the deletion. As a further verification, FACS analysis was carried out to determine CXCR4 expression (Supplementary Fig. S17).

## Reporting summary

Further information on research design is available in the Nature Portfolio Reporting Summary linked to this article.

## Data availability

All SAXS data have been deposited to the SASBDB with the following codes: SASDRH9; SASDRG9; SASDRJ9. NMR backbone assignments for CXCL12-LM used in this study are available under BMRB entry ID 52209. The experimental data that support the findings of this study are shown in the article and its supplementary materials. The Source data underlying all Figures and Supplementary are provided as Source Data file with this paper and are available at [https://doi.org/10.6084/m9.figshare.24574522]. All other data are available from the authors upon request. NMR backbone assignments of HMGB1, HMGB1-TL, CXCL12 are available under BMRB entry: ID 15149; ID 15148 and ID 16519, respectively. The NMR Coordinates of CXCL12, CXCL12-LM, and HMGB1-TL are available in the PDB with the following accession codes 2KEE; 2N55; 2YRQ, respectively. The crystallographic structure of CXCL12 with two monomers in the asymmetric unit is available in the PDB with the code 1QG7. The Alphafold model of HMGB1 is available in the AlphaFold Protein Structure Database with the following entry AF-P63159. Source data are provided with this paper.

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

## Acknowledgements

The research reported here has received funding from AIRC (Associazione Italiana per la Ricerca sul Cancro) under IG 2018 - ID. 21440 project – P.I. GM, and IG 2020 – ID. 24702 P.I. MEB, and MFAG27415 P.I RM; by Fondazione Buzzi Unicem –P.I MEB. MVM was supported by a FIRC-AIRC fellowship 24118 for Italy. LSC was supported by the RENOIR European Training Network. CZ was supported by FSHD Global Research Foundation FSHD-Winter2021-5008608933, GG has received funding from the University of Padova, Starting Grant STARS@UNIPD. The authors acknowledge the European Synchrotron Radiation Facility (ESRF) for provision of synchrotron radiation facilities under proposal number MX2386 and thank Mark Tully and Petra Pernot for assistance and support in using beamline BM29. We wish to thank Dr Joy Zhao, (Laboratory of Dynamics of Macromolecular Assembly National Institute of Biomedical Imaging and Bioengineering) for useful discussions on AUC data. Part of the present work was performed by MVM in fulfillment of the requirements for obtaining a Ph.D. degree at Vita-Salute San Raffaele University, Milan, Italy.

## Author contributions

M.V.M., C.C., C.P. produced recombinant proteins. M.V.M., F.D.L., M.G. and G.Q. analyzed NMR experiments. M.V.M., T.S and S.R. performed and analyzed ITC experiments. M.V.M. performed and analyzed AUC experiments and prepared a draft of the manuscript. G.Q. and M.V.M. performed the NMR experiments. F.D.L. and C.Z. performed and analyzed MST experiments. L.S.C., F.D.M. and M.C. performed and analyzed PLA experiments and generated Cxcr4 KO AB1 cells. F.C. performed flow cytometry. R.M. participated to the design of cell-based experiments. M.E.B. coordinated all experiments with cells and contributed to the writing of the manuscript. G.G. supervised the SAXS experiments and performed all the SAXS data analysis with the support of M.V.M., F.D.L. and M.G. M.G. generated the model of the interaction, participated to the interpretation of all the data and in manuscript preparation. G.M. supervised the study and was involved in all aspects of the experimental design, data analysis, and manuscript preparation. All authors critically reviewed the text and figures.

## Competing interests

The authors declare the following competing interests: L.S.C. is an employee and M.E.B. is founder and part-owner of HMGBiotech, a company that provides goods and services related to HMGB proteins. The remaining authors declare no competing interests.

## Additional information

**Supplementary information** The online version contains Supplementary Material available at https://doi.org/10.1038/s41467-024-45505-7.

