## [Peer Review File · Nature Communications]

The acidic intrinsically disordered region of the inflammatory mediator HMGB1 mediates fuzzy interactions with CXCL12REVIEWER COMMENTS

Reviewer #1 (Remarks to the Author):

In this manuscript, the authors describe how HMGB1 interacts with CXCL12. They combined biophysical and cell-biological approaches and found that C-terminal acidic tail is important for the HMGB1-CXCL12 interactions. Based on NMR and SAXS data, the authors conclude that HMGB1 and CXCL12 form a fuzzy chemokine heterocomplex. The concept is very interesting, and the presented data are convincing regarding the HMGB1-CXCL12 interactions. However, I think that the following major issues should be addressed for publication in Nature Communications.

1. Previously, Venereau et al. showed that CXCL12 can bind to all-thiol HMGB1 but cannot bind to disulfide HMGB1 (J. Exp. Med. 209, 1519-28 [2012]; Refs 19 in the current manuscript). The difference between all-thiol HMGB1 and disulfide HMGB1 is whether or not the Cys23-Cys45 disulfide bond in the A-box domain is formed. Unfortunately, the model presented in the current manuscript cannot explain why the redox state of the A-box domain matters for HMGB1-CXCL12 interaction. If the C-terminal acidic tail truly governs HMGB1's binding to CXCL12, why could the disulfide bond in A-box drastically diminish CXCL12-HMGB1 association?

2. Throughout the current manuscript, the redox state of HMGB1 used in the experiments is not indicated. The buffers for the experiments do not appear to include any reducing reagent. For the wild-type HMGB1 protein, the Cys23-Cys45 disulfide bond is formed in hours unless anaerobic or reducing conditions are used. How did the authors manage the redox state of HMGB1? This is an important issue because the previous work shows that the redox state of HMGB1 is crucial for CXCL12-HMGB1 interaction.

3. Table 1 indicates that the dissociation constant K_d for full-length HMGB1 at 150 mM NaCl is ~ 32 micromolar. The physiological concentrations of HMGB1 and CXCL12 are probably far lower than this K_d . If so, how could such a weak affinity be biologically relevant?

4. Regarding #3, a testable hypothesis could be that HMGB1's binding to DNA disrupt the intra-molecular interactions between the C-terminal tail and A/B-boxes and thereby enhance the tail's binding to CXCL12. This possibility might also address #1. DNA fragments are released to extracellular space through necrosis.

Reviewer #2 (Remarks to the Author):

Mantonico et al

The manuscript entitled "The acidic intrinsically disordered region of the inflammatory mediator HMGB1

mediates fuzzy interactions with chemokine CXCL12" describes the interaction between CXCL12 and HMGB1 in detail not previously explored. Specifically, they look into the contribution of the IDR to the binding. Based on the results shown in this manuscript, it is clear that the IDR plays an important role in the interaction. The diversity of techniques used to characterize this system is impressive. It should be noted that each experiment serves a purpose to the story and is not simply added on. The work in this manuscript is suitable for publication in Nature Comms, with some edits. Of note, the evidence in the manuscript clearly demonstrates that the IDR is involved in the interaction but does not clearly prove that there is a dynamic/fuzzy complex. The definition of a fuzzy complex is a complex that interacts via many different binding motifs that add to form a single complex. To demonstrate this, measurements of dynamics and exchange between binding motifs are essential. For reference, see the seminal work on Sic1 from the Forman-Kay Lab (<https://doi.org/10.1073/pnas.0809222105>) and more recently the work on ProTa/H1 from the Schuler lab (<https://doi.org/10.1038/nature2576>). This said, the conclusions in this paper do not require the complex to be dynamic.

Following are detailed comments:

The exact structures and amino acid sequences (including annotations of acidic versus basic residues that are important in the context of the manuscript) are unclear in the paper. It would be good to include a figure that is a schematic of the work that will be done.

Figure 1: Ac-pep, HMGB1-TL and HMGB1 interact with the CXCL12 dimerization surface.

Add a box-like illustration of the functional domains within CXCL12 and HMGB1 variants to visualize the difference especially between Ac-pep and HMGB1-TL and HMGB1. Highlight the sequence composition and the net charge of Ac-pep as well as the net charge of CXCL12. This likely should be added to a new figure as noted in bullet point 1.

Figure 1H CSPs (CXCL12) appear smaller compared to Figure S2 H (CXCL12-LM), but authors claim that they are similar. This should either be explained or changed.

SI Figure 1: Authors can add a few words in the main text on why CXCL12-LM mutant is locked in a monomeric state. Where are the mutations (L55C, I85C) mapped on the dimerization surface and how do they impact the monomer-multimer equilibrium?

Figure 2: The HMGB1-CXCL12 heterocomplex forms via fuzzy interaction

Figure 2A needs legends to make it clear to the readers. It's written on the figure caption but I think a box with HMGB1 in black, HMGB1 + CXCL12 in red and HMGB1 + HMGB1 TL in blue will be helpful.

Figure 2B it would be helpful to have a legend to show the subsequent addition of CXCL12 to form the complex (red) and the addition of ac-pep to compete with the complex (green).

The NMR data presented appears to be of high quality and supports the conclusion that CXCL12 interacts with HMGB1 via its acidic domain. However, I am not sure if the data in the manuscript provides enough evidence that the heterocomplex forms via fuzzy interactions because the authors don't highlight any dynamic conformations that interchange with each other unless by fuzzy they just mean to highlight the importance of the acidic IDR domain in observed binding. Further insight into the exchange dynamics between CXCL12 and HMGB1 Ac-pep could be gathered by using NMR relaxation (e.g R1, R2 hetNOE) experiments that are ideal for probing the protein dynamics and conformational flexibility on the ps-ns timescale.

Figure 3: The acidic IDR of HMGB1 interacts with CXCL12 via long-range electrostatic interactions

Figure 3B,C: It would be helpful to indicate on the figure legends the concentration of NaCl. It might be even more helpful to plot the 6 concentration-response curves (low, high salt) on the same graph to highlight the difference in the slope of the curves. My interpretation from the text is that binding parameters cannot be determined in the presence of 150mM NaCl, but the curves shown seem to include fits. If they are fit, the parameters should be shown. If not, this needs to be clarified.

Overall, ITC and MST results show that the acidic IDR of HMGB1 interacts with CXCL12. To strengthen the conclusion that the binding occurs via electrostatic interactions, one could generate different acidic IDR variants by mutating the charged residues within the acidic region of 186-214 that are hypothesized to engage in key interactions. What about an HMGB1 arginine-rich tail construct mentioned in the discussion to be associated with aberrant phase separation?

Figure 4: HMGB1 and CXCL12 form an equimolar heterocomplex

Figure 4C: Explain on the caption what the arrow indicates

Overall, I think the modeling of the AUC data needs to be clarified and it should be explained why the model used, and not other binding models, is correct. This could be done by fitting to multiple different models.

Figure 5: SAXS studies of free HMGB1, CXCL12 and of HMGB1-CXCL12

The SAXS data appears to be of high quality, but it is hard to determine this without seeing the whole concentration series before scaling and merging. The concentration series should be included in the supplementary information.

A molar ratio of 1:2 is used to define the complex. It is a bit shocking that this works as well as it does. Small changes in flexible regions of proteins, for example, can shift the SAXS profile. The presence of free proteins in solution makes a lot of the docking and fitting a bit suspect. Figure S5 shows that at higher molar ratios the MW is no longer accurate, but this is not a high enough standard to show that the scattering profile is pure complex. The correct way to fit this data would be to include the known SAXS profiles of each monomer in the fit. The profile observed will be a linear combination of monomers and dimer (assuming the complex is 1:1)

It is not clear why two different docking methods are used.

It is also important to note that these data, which appear to show a very strong dimeric complex and the slow exchange in the NMR experiments are not aligned with the notion of a fuzzy complex. This should be clarified in the text. It is possible I am misinterpreting the results.

Figure 6: The acidic IDR modulates HMGB1-CXCL12 binding to CXCR4 on AB1 cells

Figure 6A,B: Scale bars on panel A?

Would it be reasonable to ask for the number of red spots on images presented in B? Probably not. What about \log_{10} intensity of the pixels?

What are the key residues in transmembrane CXCR4 helix that mediate binding to the heterocomplex?

Can you support these data with a biochemical assay that measures ternary complex formation?

Figure 7: The model

Top: How were these ensembles generated?

Bottom: The acidic IDR coloring is not clear to me (I can't seem to recognize the highlighted spheres)

Section-by-section comments:

Introduction

Add a few more words on the structural characteristics of CXCL12 and describe the sites of the dimerization surface. (as noted in the figure comments. A figure could be included to help with this)

Add a paragraph and a few more references on the disease-related consequences of CXCL12-HMGB1 interactions.

Provide more information on the importance of HMGB1-CXCL12 heterocomplex on CXCR4 receptor's biological function.

Add a few words on the importance of fuzzy interactions in protein-protein interactions. The literature cited on this topic is a bit thin. If demonstrating a fuzzy complex is important to the authors they should define precisely what they mean by this and what evidence will be presented to prove it.

Results

See figure-by-figure comments

Discussion

HMGB1's propensity to phase separate would be helpful if discussed also in the introduction.

Add more references on the importance of fuzzy interactions on the formation of condensates and discrete complexes (e.g PMID: 36416859).

Discuss in more details previous NMR studies on this complex. Previous work often required looking at other citations to understand the current work.

Discuss in more details the therapeutic value of targeting the heterocomplex interactions.

Reviewer #3 (Remarks to the Author):

In their manuscript, authors studied in detail the molecular interaction (by NMR, ITC and MST) between the chemokine CXCL12 and HMGB1 either with or without its negatively charged intrinsically disordered region (IDR). In addition, they synthesized the IDR domain and evaluated the effect of this domain as such. They prove the importance of the IDR and long-range electrostatic interactions for the low μM affinity heterocomplex interaction between HMGB1 with CXCL12. In contrast to previous reports, authors show that HMGB1 forms an equimolar "fuzzy" 1:1 complex with CXCL12. In addition, they show that acidic IDR of HMGB1 facilitates the binding of the heterocomplex to the chemokine receptor CXCR4.
Comments:

1. Is the effect of the IDR comparable to the effect of negatively charged glycosaminoglycans such as

heparan sulfate? Does addition of heparan sulfate have the same effect as addition of the IDR? Is HMGB1 or its IDR releasing CXCL12 from GAGs improving its interaction with CXCR4?

2. Reduced and oxidized forms of HMGB1 have a different activity. Can authors indicate which form of HMGB1 is used in this study and is reduction/oxidation affecting the interactions and the effects of the IDR?

Point-to-Point reply to the reviewers

Reviewer #1 (Remarks to the Author):

In this manuscript, the authors describe how HMGB1 interacts with CXCL12. They combined biophysical and cell-biological approaches and found that C-terminal acidic tail is important for the HMGB1-CXCL12 interactions. Based on NMR and SAXS data, the authors conclude that HMGB1 and CXCL12 form a fuzzy chemokine heterocomplex. The concept is very interesting, and the presented data are convincing regarding the HMGB1-CXCL12 interactions. However, I think that the following major issues should be addressed for publication in Nature Communications.

1. Previously, Venereau et al. showed that CXCL12 can bind to all-thiol HMGB1 but cannot bind to disulfide HMGB1 (J. Exp. Med. 209, 1519-28 [2012]; Refs 19 in the current manuscript). The difference between all-thiol HMGB1 and disulfide HMGB1 is whether or not the Cys23-Cys45 disulfide bond in the A-box domain is formed. Unfortunately, the model presented in the current manuscript cannot explain why the redox state of the A-box domain matters for HMGB1-CXCL12 interaction. If the C-terminal acidic tail truly governs HMGB1's binding to CXCL12, why could the disulfide bond in A-box drastically diminish CXCL12-HMGB1 association?

We thank the reviewer for this comment, that has prompted us to perform additional experiments to clarify this point. Overall, our data show that **both** the structured domains, in particular BoxA, **and** the acidic IDR contribute to complex formation, in line with the multivalent nature of this interaction. Herein the acidic IDR plays a dual role: on the one hand it works like an antenna effectively recruiting CXCL12 via long range electrostatic interactions, on the other hand, as properly pointed out by this reviewer, this binding weakens HMGB1 intramolecular interaction, and shifts HMGB1 conformational equilibrium towards a more open conformation, thus making BoxA more accessible to the interaction with CXCL12. The fact that the acidic IDR is important for the interaction does not imply that the binding to CXCL12 occurs only through the acidic IDR. In fact, NMR titrations of fully reduced tailless HMGB1 (frHMGB1-TL) with CXCL12 show that the binding occurs even in the absence of the acidic IDR, with a preferential binding on BoxA (Figure 3 of the current manuscript).

To clarify this point in the revised manuscript we have performed additional binding experiments using disulfide HMGB1 (dsHMGB1-TL and dsHMGB1) as control. As a matter of fact, NMR titrations performed on ¹⁵N dsHMGB1-TL with CXCL12 show reduced intensity changes on BoxA as compared to frHMGB1-TL (**Supplementary Figure S4a and Figure 3**), in line with the important role of BoxA and of its oxidation status. Similarly, also the reversed titration performed on ¹⁵N CXCL12 with dsHMGB1-TL highlights reduced spectral perturbations as compared to the one with frHMGB1-TL (**Supplementary Figure S4b and Figure 2**). However, it is important to note that in the presence of the acidic IDR the binding of dsHMGB1 is not abrogated, as shown by NMR titrations of ¹⁵N CXCL12 with full length dsHMGB1 and *vice versa* (**Supplementary Figure S4c-d**). Herein, in ¹⁵N CXCL12 HSQC the spectral perturbations are due to the interaction with the acidic IDR only (and not with BoxA). In ¹⁵N dsHMGB1 HSQC experiments, spectral perturbations (as in frHMGB1) are mainly due to the different intramolecular interactions of the acidic IDR occurring upon interaction with CXCL12. Along the same line MST titrations of CXCL12 with dsHMGB1 and frHMGB1 clearly highlight the difference in binding between the two HMGB1 redox forms, showing that dsHMGB1 interacts one order of magnitude less with respect to frHMGB1 (**Supplementary Figure S4e**).

Taken together, these data support the idea that both the acidic IDR and BoxA (and its redox state) play crucial roles in the interaction with CXCL12. The apparent inconsistency with Venereau's study can be reconciled by the fact that in that study the physical interaction between CXCL12 and the different HMGB1 redox forms was examined through a biochemical assay (hybrid ELISA), while in our study we took advantage of biophysical methods (NMR, MST) which can detect also weak binding events in solution (in particular NMR, up to mM Kd values). Overall, our approach appears well suited for describing the multivalent interactions occurring in this highly dynamic system, as it also allows us to detect differences in affinity between frHMGB1 and dsHMGB1, which might not have been evident in a simple, low-resolution "yes-or-no" system like ELISA.

2. Throughout the current manuscript, the redox state of HMGB1 used in the experiments is not indicated. The buffers for the experiments do not appear to include any reducing reagent. For the wild-type HMGB1 protein, the Cys23-Cys45 disulfide bond is formed in hours unless anaerobic or reducing conditions are used. How did the authors manage the redox state of HMGB1? This is an important issue because the previous work shows that the redox state of HMGB1 is crucial for CXCL12-HMGB1 interaction.

We thank all the reviewers for having pointed out the lack of clarity on the oxidation state of HMGB1 used in the manuscript. Throughout the manuscript we have used the fully reduced form of HMGB1 (and constructs thereof). In the revised version, for comparison, we have also used the oxidized form. For sake of clarity we define now the fully reduced form(s) as frHMGB1 (frHMGB1-TL) and the disulphide form(s) as dsHMGB1 (dsHMGB1-TL). The heterocomplex with CXCL12 has been now defined as frHMGB1•CXCL12.

We are aware of the fact that Cys23 and Cys45 easily form a disulphide bond in non reducing conditions. Of note, by NMR it is possible to verify the oxidation status of the sample, as the aliphatic region of dsHMGB1 and frHMGB1 are readily distinguishable by 1D ¹H NMR (**Figure reviewer only 1**). Thus, as a standard procedure in our lab, before and after NMR, ITC, MST, SAXS, AUC experiments we always check by NMR the oxidation status of HMGB1 (stock solutions and/or samples directly used). In our conditions (buffer, ionic strength, temperature) and for the duration of the experiments (a few minutes up to a few hours) HMGB1 maintained its reduced form even in the absence of DTT.

Figure reviewer only 1: Zoom into the aliphatic region of frHMGB1 along time. Mono-dimensional spectra of ^{15}N frHMGB1 in non-reducing conditions (20 mM phosphate buffer, 20 mM NaCl, T=298K) along time. The methyl resonances of V35 and V19, that are close in space to C22 and C44, are *spy-signals* for the oxidation status of BoxA. This time course experiments indicates that frHMGB1 maintains its reduced form in non-reducing conditions for at least 12 hours.

3. Table 1 indicates that the dissociation constant K_d for full-length HMGB1 at 150 mM NaCl is ~ 32 micromolar. The physiological concentrations of HMGB1 and CXCL12 are probably far lower than this K_d . If so, how could such a weak affinity be biologically relevant?

We agree with the reviewer that the *in vitro* affinity in solution between HMGB1 and CXCL12 is relatively low, compared to the extracellular concentrations of both components. We cannot exclude that additional effects, like the interaction with a third partner like DNA (see further) might increase the affinity. As the interaction between CXCL12 and frHMGB1 occurs in the extracellular milieu, most likely at the cell membrane, we cannot exclude that locally the concentrations of both HMGB1 and CXCL12 might be higher, thus favouring the interaction. Additional mechanisms and cooperative binding phenomena among multiple actors at the cell surface might also occur, such as reduced dimensionality, hindered diffusion and/or avidity^{2,3}, herewith synergistically and/or allosterically increasing the affinity in the extra-cellular environment. Along this line, our PLA experiments show that the heterocomplex forms on the cell surface when CXCR4 is expressed (see **Figure 8**), thus supporting the notion that additional binding events, such as the interaction with CXCR4, can favour complex formation at physiological concentrations.

It is also important to note that the dissociation constants measured for chemokine heterocomplexes differ, according to the methods applied. In general, solution methods (like NMR and MST) measure always weaker affinities as compared to methods, like SPR, that use surface-immobilized binding partners, that might better mimic binding phenomena occurring on the membrane surface^{4,5}.

4. Regarding #3, a testable hypothesis could be that HMGB1's binding to DNA disrupt the intramolecular interactions between the C-terminal tail and A/B-boxes and thereby enhance the tail's binding to CXCL12. This possibility might also address #1. DNA fragments are released to extracellular space through necrosis.

We thank the reviewer for this very inspiring suggestion, that prompted us to investigate the possible role of DNA in complex formation. To this aim we chose a CpG-DNA oligonucleotide sequence (DNA1018), that is known to bind HMGB1 and activate dendritic cells and macrophages to produce inflammatory cytokines¹.

During NMR titrations we observed that CXCL12, in agreement with its basic surface, strongly interacts with CpG oligonucleotides, as assessed by strong line broadening effects in NMR titrations of ^{15}N CXCL12 with 1018 (**Figure reviewer only 2a**). The interaction of CXCL12 with DNA is in line with previous reports describing the formation of nanoparticles between CXCL12 and DNA⁶.

Overall, interpretation of CXCL12 titration experiments (MST or NMR) performed in the presence of both CpG oligonucleotides and frHMGB1 was extremely difficult and ambiguous (**Figure reviewer only 2a,b**). The multiple equilibria occurring in solution caused extreme line broadening effects in NMR titrations, thus hampering further analysis. Moreover, MST experiments performed adding to fluorescently labeled CXCL12 a preformed 1:1 frHMGB1:DNA1018 complex showed only a modest increase in the affinity, which remained in the micromolar range (**Figure for reviewer**

only 2c). Still, this hypothesis remains extremely fascinating and deserves dedicated attention in future biophysical experiments.

Figure reviewer only 2: **a)** ^1H - ^{15}N HSQC spectra of 0.1 mM ^{15}N CXCL12 free (left), in the presence of 0.2 mM DNA1018 (middle), in the presence of 0.2 mM DNA1018 and 0.1 mM frHMGB1 (right). **b)** ^1H - ^{15}N HSQC spectra of 0.1 mM ^{15}N frHMGB1 free (left), in the presence of 0.1 mM DNA1018 (middle), in the presence of 0.1 mM DNA1018 and 0.1 mM CXCL12(right), pH 6.3, 20 mM $\text{NaH}_2\text{PO}_4/\text{Na}_2\text{HPO}_4$, 150 mM NaCl. **c)** normalized variation of MST signal of labeled CXCL12 in the presence of frHMGB1 (black, K_d $31.8 \pm 1.4 \mu\text{M}$) and of frHMGB1:DNA1018 (orange, $16 \pm 1.4 \mu\text{M}$).

Reviewer #2 (Remarks to the Author):

Mantonico et al

The manuscript entitled "The acidic intrinsically disordered region of the inflammatory mediator HMGB1 mediates fuzzy interactions with chemokine CXCL12" describes the interaction between CXCL12 and HMGB1 in detail not previously explored. Specifically, they look into the contribution of the IDR to the binding. Based on the results shown in this manuscript, it is clear that the IDR plays an important role in the interaction. The diversity of techniques used to characterize this system is impressive. It should be noted that each experiment serves a purpose to the story and is not simply added on. The work in this manuscript is suitable for publication in Nature Comms, with some edits. Of note, the evidence in the manuscript clearly demonstrates that the IDR is involved

in the interaction but does not clearly prove that there is a dynamic/fuzzy complex. The definition of a fuzzy complex is a complex that interacts via many different binding motifs that add to form a single complex. To demonstrate this, measurements of dynamics and exchange between binding motifs are essential. For reference, see the seminal work on Sic1 from the Forman-Kay Lab (<https://doi.org/10.1073/pnas.0809222105>) and more recently the work on ProTa/H1 from the Schuler lab (<https://doi.org/10.1038/nature2576>). This said, the conclusions in this paper do not require the complex to be dynamic.

We thank the reviewer for the overall positive evaluation of our manuscript. We have highly appreciated the useful suggestions/comments that we think have contributed to improve the quality of our manuscript. We have tried to address the reviewer's issues and requests, as described in the reply to the detailed comments.

Following are detailed comments:

The exact structures and amino acid sequences (including annotations of acidic versus basic residues that are important in the context of the manuscript) are unclear in the paper. It would be good to include a figure that is a schematic of the work that will be done.

Figure 1: Ac-pep, HMGB1-TL and HMBGB1 interact with the CXCL12 dimerization surface.

Add a box-like illustration of the functional domains within CXCL12 and HMGB1 variants to visualize the difference especially between Ac-pep and HMGB1-TL and HMGB1. Highlight the sequence composition and the net charge of Ac-pep as well as the net charge of CXCL12. This likely should be added to a new figure as noted in bullet point 1.

We thank the reviewer for the suggestion, we have now included an additional figure reporting the amino acid sequences, the basic and acidic residues and the charges of the constructs used in the manuscript (**now Figure 1**).

Figure 1H CSPs (CXCL12) appear smaller compared to Figure S2 H (CXCL12-LM), but authors claim that they are similar. This should either be explained or changed.

We thank the reviewer for this observation. Indeed, the CSPs observed in CXCL12-LM titrations in terms of profile are similar to the wild-type one, but their entity is higher as compared to the wild-type protein. The main purpose of conducting titrations with the mutant protein was to differentiate CSPs resulting from dimerization phenomena from those arising from interactions with frHMGB1 and its constructs. Our results suggest that the CSPs observed are primarily a consequence of the interaction with frHMGB1, rather than oligomerization. To avoid any confusion, we have changed the sentence, as suggested by the reviewer (page 6-7).

SI Figure 1: Authors can add a few words in the main text on why CXCL12-LM mutant is locked in a monomeric state. Where are the mutations (L55C, I85C) mapped on the dimerization surface and how do they impact the monomer-multimer equilibrium?

We have clarified in the text that we used a CXCL12 variant that removes the confounding effects of CSPs originating from shifts in the dimer-monomer equilibrium. We have also added figures (**Figure 1a, Supplementary Figure S1**) showing the location of the mutations. This mutant has been generated and described in detail in a publication by Volkman and coworkers⁷. It blocks the angle between helix $\alpha 1$ in $\beta 1$ herewith hampering dimerization. We have briefly described the

effect of these mutations in the text (page 6). The mutations are not in the dimerization surface (**Supplementary Figure S1**).

Figure 2: The HMGB1-CXCL12 heterocomplex forms via fuzzy interaction

Figure 2A needs legends to make it clear to the readers. It's written on the figure caption but I think a box with HMGB1 in black, HMGB1 + CXCL12 in red and HMGB1 + HMGB1 TL in blue will be helpful.

Following the reviewer's advice we have added a legend inside the figure, for sake of space we had to remove one panel from the original figure (**now Figure 3**).

Figure 2B it would be helpful to have a legend to show the subsequent addition of CXCL12 to form the complex (red) and the addition of ac-pep to compete with the complex (green).

Following the reviewer's advice we have added a legend inside the figure (**now Figure 3**)

The NMR data presented appears to be of high quality and supports the conclusion that CXCL12 interacts with HMGB1 via its acidic domain. However, I am not sure if the data in the manuscript provides enough evidence that the heterocomplex forms via fuzzy interactions because the authors don't highlight any dynamic conformations that interchange with each other unless by fuzzy they just mean to highlight the importance of the acidic IDR domain in observed binding. Further insight into the exchange dynamics between CXCL12 and HMGB1 Ac-pep could be gathered by using NMR relaxation (e.g R1, R2 hetNOE) experiments that are ideal for probing the protein dynamics and conformational flexibility on the ps-ns timescale.

We thank the reviewer for this comment, which has prompted us to perform additional experiments to clarify this point. According to the concept of fuzzy interaction, as defined by Tompa and Fuxreiter⁸, fuzzy complexes are those complexes in which at least one of the elements in the complex remains dynamic, thus it cannot be properly described by a defined structure, but has the characteristics of an heterogeneous ensemble. Our data strongly support an interaction mechanism in which a significant degree of disorder is maintained in the frHMGB1•CXCL12 complex. The sources of disorder/multivalency, supporting the formation of fuzzy interactions are manifold: i. HMGB1, which has been previously defined as a "fuzzy" protein⁹, maintains its intrinsic flexibility in its interaction with CXCL12; ii. CXCL12 establishes with the same interaction surface multivalent interactions with BoxA and the acidic IDR; iii. the Ac-pep maintains its conformational flexibility when bound to CXCL12 as shown by relaxation experiments; iv. the Ac-pep contains multivalent sites able to interact with CXCL12.

The features described in point iii) and iv) emerge from additional experiments performed in the revised manuscript and described in the following:

Following the reviewer's suggestion, to get further insights into the dynamics of Ac-pep upon interaction with CXCL12, we have cloned, expressed and purified a recombinant labeled (¹⁵N and ¹⁵N/¹³C) peptide corresponding to the acidic IDR (Ac-pep_{rec}) and performed NMR titrations with unlabelled CXCL12. The ¹H-¹⁵N HSQC spectrum of free ¹⁵N Ac-pep_{rec} has the characteristic of an intrinsically disordered domain with reduced peak dispersion (both on ¹H and ¹⁵N chemical shifts) and high signal overlap. As typically observed in fuzzy interactions, these characteristics are maintained upon addition of equimolar concentration of unlabelled CXCL12, with extremely small chemical shift perturbations due to the average chemical environment of the corresponding residues. The appearance in the aliphatic regions of CXCL12 methyls in the bound position

confirms complex formation (**Figure reviewer only 3**). The pronounced overlap in the NMR spectra and the repetitive amino acid sequence of ^{15}N -Ac-pep_{rec} precluded residue-specific assignments. The clusters of H α -C α and H β -C β peaks of the Aspartic and Glutamic residues observed in the ^1H - ^{13}C HSQC did not exhibit detectable chemical shift perturbations upon titration with CXCL12 and additional resonances did not emerge (**Figure 4a-d**). This indicates that the Ac-pep_{rec} maintains a high degree of disorder upon binding, without the induction of persistent or transiently populated secondary structures. Hence, the lack of structure formation in the complex implies great flexibility and suggests a highly dynamic interconversion within a large ensemble of configurations and relative arrangements of the acidic IDR.

Both free and bound states exhibited negative heteronuclear NOE values, signifying high flexibility on the picosecond to nanosecond timescale. A modest increase in complex heteronuclear NOE values indicated a slight reduction in peptide mobility in the presence of CXCL12. The increase in R2/R1 and reduced peak intensity was in line with complex formation, with the consequent slowdown of its tumbling in solution and with the dynamic exchange between multiple CXCL12 binding sites (**Figure 4e-f**).

Finally, the presence of multiple interchangeable binding sites within the acidic IDR was further confirmed by titrating ^{15}N CXCL12 with different fragments from the acidic IDR (Ac-pep₁₈₅₋₁₉₅ and Ac-pep₂₀₄₋₂₁₄), that akin to Ac-pep, targeted the CXCL12 dimerization surface (**Supplementary Figure S5a-f**). Thus, within the acidic IDR multiple interchangeable binding sites exist that can dynamically interact with CXCL12. These results can be found on page 7-8. Unfortunately, the company who provided us with the synthetic peptides has not been able to synthesize the peptide corresponding to residues 195-204, thus we could not perform the corresponding titrations. However, we expect that this peptide would have given similar results as the other fragments.

Taken together, we think that with these additional experiments we provided evidence that within the frHMGB1•CXCL12 heterocomplex fuzzy interactions do occur.

Figure reviewer only 3: ^1H NMR spectra of the aliphatic region of free ^{15}N Ac-pep_{rec} (black, 0.1mM), of ^{15}N Ac-pep_{rec} (0.1 mM) with equimolar amount of CXCL12 (red), ^{15}N CXCL12 (0.1 mM) with equimolar amount of Ac-pep (purple), free ^{15}N CXCL12 (0.1 mM) (green). The red and the purple spectra indicate that the complex is formed.

Figure 3: The acidic IDR of HMGB1 interacts with CXCL12 via long-range electrostatic interactions

Figure 3B,C: It would be helpful to indicated on the figure legends the concentration of NaCl. It might be even more helpful to plot the 6 concentration-response curves (low, high salt) on the same graph to highlight the difference in the slope of the curves. My interpretation from the text is that binding parameters cannot be determined in the presence of 150mM NaCl, but the curves shown seem to include fits. If they are fit, the parameters should be shown. If not, this needs to be clarified.

We have included in the figures of the MST traces the NaCl concentrations (**now Figure 5**), for sake of clarity we have preferred to maintain separated the two salt conditions.

We apologize for any confusion regarding the fitting of ITC and MST data. While it was possible to fit the MST curves at 150 mM NaCl, it was not possible to fit the ITC data at high ionic strength (**Supplementary Figure S6c,d**) and in ITC titration with frHMGB1-TL (**Figure 5c**). We corrected the legend of Figure 5 specifying that only ITC titrations with Ac-pep and frHMGB1 could be fitted. The parameters resulting from the titrations that could be fitted are summarized in Table 1 as indicated in the legend.

Overall, ITC and MST results show that the acidic IDR of HMGB1 interacts with CXCL12. To strengthen the conclusion that the binding occurs via electrostatic interactions, one could generate different acidic IDR variants by mutating the charged residues within the acidic region of 186-214 that are hypothesized to engage in key interactions. What about an HMGB1 arginine-rich tail construct mentioned in the discussion to be associate with aberrant phase separation?

Following the reviewer's advice, we have performed a titration of ¹⁵N CXCL12 with the HMGB1 arginine/lysine-rich tail construct (RKMRKMKRMRRRRKMKMKMKKKMKKMNK) corresponding to the aberrant form of HMGB1 leading to a human malformation syndrome. This arginine-rich tail, as expected, does not interact with CXCL12, thus corroborating the conclusion that electrostatic interactions contribute to the binding. These results are now shown in **Supplementary Figure S5g-i** and described on page 9.

Figure 4: HMGB1 and CXCL12 form an equimolar heterocomplex

Figure 4C: Explain on the caption what the arrow indicates

The arrow indicates the SV-peak displacement upon addition of increasing concentrations of CXCL12, for sake of clarity we have removed it from the figure (**now Figure 6**).

Overall, I think the modeling of the AUC data needs to be clarified and it should be explained why the model used, and not other binding models, is correct. This could be done by fitting to multiple different models.

Following the reviewer's suggestion we added more details on the analysis of the AUC data. In the methods section we have clearly stated that, in AUC analysis, the data are not fitted a priori to a binding stoichiometry model (e.g. 1:1 or 1:2) and no binding model assumption is made. In detail, the experimental sedimentation data, whose physical observable is absorbance, are fitted with mathematical equations describing the sedimentation process without assuming any binding stoichiometry. The sedimentation process is described as a superposition of normalized Lamm equation solutions of ideally sedimenting species at a range of sedimentation coefficients $s(1 \dots N)$, with N indicating the range of molecules size expressed as sedimentation coefficient in Svedberg and using the hydrodynamic scaling law to estimate the corresponding diffusion coefficients. The Lamm equation solutions, once superimposed, give the number of species, molecular weight and

shape. Even though we used an excess of CXCL12 which could have been sufficient to generate a 1:2 complex, results deriving from the Lamm equation superimposition gave a 1:1 stoichiometry.

Figure 5: SAXS studies of free HMGB1, CXCL12 and of HMGB1-CXCL12

The SAXS data appears to be of high quality, but it is hard to determine this without seeing the whole concentration series before scaling and merging. The concentration series should be included in the supplementary information.

In response to the reviewer's request, we have incorporated the spectra from the dilution series of individual proteins, namely frHMGB1 and CXCL12, as well as the results of the titration experiments. These data are now presented in the Supplementary Materials (**Supplementary Figures S9, S10 and S12**). The SAXS data can be accessed in the Source Data section accompanying this paper.

A molar ratio of 1:2 is used to define the complex. It is a bit shocking that this works as well as it does. Small changes in flexible regions of proteins, for example, can shift the SAXS profile. The presence of free proteins in solution makes a lot of the docking and fitting a bit suspect.

Figure S5 shows that at higher molar ratios the MW is no longer accurate, but this is not a high enough standard to show that the scattering profile is pure complex. The correct way to fit this data would be to include the known SAXS profiles of each monomer in the fit. The profile observed will be a linear combination of monomers and dimer (assuming the complex is 1:1).

We concur with the reviewer's observation that subtle alterations in flexible regions can indeed influence the SAXS profile. However, in this particular context the following factors should be considered: our system comprises a small globular protein, CXCL12 (9 kDa), which dynamically interacts with a larger one, frHMGB1 (24.7 kDa), that has several regions of intrinsic flexibility and whose scattering signal largely dominates over the smaller molecule. Thus, during the titration series (at least up to 1:2) the overall scattering signal primarily stems from the biggest molecules, i.e.: free frHMGB1 (24.7 kDa) and frHMGB1•CXCL12 (33.7 kDa). In our experimental conditions (assuming a K_d of 2 μM , as determined by ITC and MST and a 1:1 stoichiometry, as indicated by AUC), in the presence of two CXCL12 equivalents frHMGB1 is almost fully saturated, and the concentration of free frHMGB1 (2 μM) is almost negligible with respect to the frHMGB1•CXCL12 heterocomplex (76 μM). Therefore, in this scenario, the contribution to the scattering curves of the free frHMGB1 becomes nearly negligible with respect to the heterocomplex, and the resulting molecular weight (MW) aligns with the formation of a 1:1 complex. At frHMGB1:CXCL12 1:2 also free CXCL12 (that is much smaller in size) is expected to have a negligible contribution to the scattering curve. Conversely, at higher molar ratio the contribution to the SAXS curve of CXCL12 is no longer negligible, and the molecular weights do not align with the formation of the complex. In this sense we fully agree with reviewer2's observation that at higher molar ratios (1:4; 1:6), the accuracy of the molecular weight determination is compromised. This is precisely the reason why we chose to exclude these curves from our analysis and to focus on the 1:2 SAXS data. These considerations have been added in the manuscript (page 12), and in the notes of Table 2 we specified that the SAXS parameters have been obtained from the analysis of the SAXS curve of HMGB1 in the presence of 2 equivalents of CXCL12.

In response to the reviewer's valuable suggestion to perform a linear combination of the free components and the complex to fit the SAXS data, we would like to emphasize the extreme challenges posed by this system. They arise from the inherent dynamics of the largest protein, frHMGB1, and from its various modes of binding to CXCL12. Consequently, a straightforward

linear combination of the components is difficult. Nevertheless, in response to the reviewer's suggestion to fit the SAXS data, we utilized OLIGOMER¹⁰ from the ATSAS package to estimate the volume fractions of the individual components contributing to the overall scattering signal. OLIGOMER uses form factors to fit experimental scattering data from multi-component protein mixtures. On turn, form-factors can be derived from known structures or from dummy atoms models that can be computed using the ATSAS program CRY SOL. In our case, considering the flexibility of both frHMGB1 and frHMGB1•CXCL12 we used the dummy atoms envelopes. Both the free frHMGB1 structure (one representative structure from EOM) and the best docking poses obtained with SASREF and FoXSDock fit relatively well in the envelopes obtained and are in line with the experimental Rg and Dmax (**Figure reviewer only 4b**).

According to OLIGOMER analysis, when frHMGB1 is titrated with one equivalent of CXCL12 an equimolar frHMGB1•CXCL12 heterocomplex is formed representing about 80% of the total volume fraction, alongside a non-negligible fraction of free frHMGB1 in solution (approximately 18%), which contributes to the overall scattering curve. In this condition the volume fraction of CXCL12 appears negligible. In the presence of two equivalents of CXCL12, the equimolar heterocomplex constitutes the predominant volume fraction within the SAXS spectrum, with virtually no contribution of free frHMGB1 and CXCL12 to the volume fraction (**Figure reviewer only 4b**). Conversely, at 1:4 and 1:6 ratios OLIGOMER detects an increased presence of free CXCL12, approximately representing 7 % and 18 % of the total volume fractions, respectively. Taken together, the OLIGOMER estimation of the contributions to the scattering curves of the different components supports our choice to analyse the 1:2 condition as the one that best represents the scattering behaviour of frHMGB1•CXCL12 in solution.

Figure reviewer only 4: Estimation of unbound frHMGB1 and CXCL12 during complex formation. (a) Fitting with OLIGOMER (red lines, with indicated χ^2 values) to the experimental SAXS data from 1:1 HMGB1: CXCL12 up to 1:6 molar ratio. **(b)** Estimation of fraction components within the SAXS data from 1:1 to 1:6 conditions with the contributions of the equimolar frHMGB1•CXCL12 heterocomplex (magenta), free frHMGB1 (grey) and free CXCL12 (green). The best docking complex poses obtained with SASREF and FoXSDock, frHMGB1 structure (one representative structure from EOM), and CXCL12 (pdb: 2kee) have been fitted in their corresponding dummy atoms envelopes.

It is not clear why two different docking methods are used.

We opted to employ two distinct docking methods to effectively leverage our experimental (NMR) data. On one hand, our dissecting approach has shown that the interaction involves specific surfaces, comprising residues 15-44 on BoxA and residues 23-28, 66-67 on CXCL12. This information can be exploited taking advantage of the SASREF method, a rigid body docking approach guided in our case by information from NMR experiments, which proved to be well-suited for generating initial models to be utilized in subsequent EOM calculations.

On the other hand, although the interaction surface of ¹⁵N CXCL12 with the acidic IDR can be easily identified (again involving residues 23-28, 66-67), the intrinsically disordered nature of the acidic IDR together with its highly repetitive sequence prevents the identification by NMR of a well-defined interaction surface on the acidic IDR. Thus, in the absence of experimental data specifying the residues of the acidic IDR primarily involved in the interaction with CXCL12, FoXSDock appears to be a more suitable choice than SASREF for generating plausible initial models that account for the various binding modes of CXCL12 with the entire acidic IDR. Specifically, FoXSDock generated multiple CXCL12 binding poses along the acid IDR and the best model in terms of χ^2 value was chosen. These models were then used for subsequent EOM calculations. We have added few sentences in the results to explain why we used two docking programs (page 12).

It is also important to note that these data, which appear to show a very strong dimeric complex and the slow exchange in the NMR experiments are not aligned with the notion of a fuzzy complex. This should be clarified in the text. It is possible I am misinterpreting the results.

We apologize for any confusion on the strength of frHMGB1/CXCL12 interaction.

Our NMR experiments on CXCL12 and full-length frHMGB1 (and constructs thereof) reveal that the complex formation takes place within the intermediate exchange regime on the NMR timescale. This is evident from the significant line broadening effects observed in the NMR titrations. This broadening phenomenon is consistent with the micromolar affinities determined by ITC and MST measurements, as well as with the presence of multiple equilibria occurring in the solution. We have clarified in the text that the interaction occurs in the intermediate exchange regime on the NMR timescale (see in NMR methods and on page 5).

Figure 6: The acidic IDR modulates HMGB1-CXCL12 binding to CXCR4 on AB1 cells

Figure 6A,B: Scale bars on panel A?

We added the scale bars in the figure (**now Figure 8**)

Would it be reasonable to ask for the number of red spots on images presented in B? Probably not. What about log₁₀ intensity of the pixels?

Indeed in **Figure 8b** data were expressed as the average number of PLA dots per cell. In **Figure 8a,c**, where the number of dots was difficult to count due to overlaps, data were expressed as Raw Integrated Density (sum of all pixel intensities in region of interest, in arbitrary units) normalized to the area of the respective cytosolic regions. This information is now reported in the methods section.

What are the key residues in transmembrane CXCR4 helix that mediate binding to the heterocomplex?

To precisely answer to this question, an experimental three-dimensional structure of the ternary complex is required. The ternary complex shown in **Figure 9** has only an illustrative purpose. The model derives from the superposition of one representative complex structure (derived from EOM) onto the 3D-model of CXCL12 in complex with CXCR4, previously published¹¹ (at present a CXCL12-CXCR4 high resolution structure is not available). Thus, in our illustrative model the CXCR4 transmembrane helix residues that mediate binding to the heterocomplex are the same that mediate binding to CXCL12 alone, i.e. to residues located on its N-terminal region. Notably, this region, according to our NMR titrations, does not appear to be involved in the interaction with frHMGB1. At this stage, we do not know whether binding of the heterocomplex to CXCR4 induces conformational changes on the receptor, thus favoring different intermolecular contacts with CXCL12 residues and possibly also with HMGB1 on the external loops. This and related other questions will be object of future studies.

Can you support these data with a biochemical assay that measures ternary complex formation?

To support our PLA data showing the formation of the heterocomplex on the cells surface via interaction with CXCR4, we have added a control experiment in which AB1 cells knocked out for CXCR4 were used. In this case we do not observe the presence of the heterocomplex on the surface in PLA experiments, thus supporting the existence of a ternary complex composed by frHMGB1•CXCL12•CXCR4 (**Figure 8b**).

Figure 7: The model

Top: How were these ensembles generated?

The ensembles are representative structures derived from EOM calculations, as specified now in the legend.

Bottom: The acidic IDR coloring is not clear to me (I can't seem to recognize the highlighted spheres)

We have changed the acidic IDR coloring and we have removed the spheres (**now Figure 9**).

Section-by-section comments:

Introduction

Add a few more words on the structural characteristics of CXCL12 and describe the sites of the dimerization surface. (as noted in the figure comments. A figure could be included to help with this)

We have now added in the introduction few more words on the structural characteristics of CXCL12 and a figure (**Figure 1**). We have also added a supplementary figure highlighting the dimerization surface (**Supplementary Figure S1a,b**).

Add a paragraph and a few more references on the disease-related consequences of CXCL12-HMGB1 interactions.

We have added a sentence quoting an example of inflammation related disease (malignant mesothelioma) involving the HMGB1-CXCL12-CXCR4 axis (page 4).

Provide more information on the importance of HMGB1-CXCL12 heterocomplex on CXCR4 receptor's biological function.

We have added some more information on HMGB1-CXCL12-CXCR4 axis (page 4).

Add a few words on the importance of fuzzy interactions in protein-protein interactions. The literature cited on this topic is a bit thin. If demonstrating a fuzzy complex is important to the authors they should define precisely what they mean by this and what evidence will be presented to prove it.

Following the reviewer's suggestion we have expanded the description of fuzzy interactions both in the introduction (last paragraph) and in the discussion (page 14). We have added some more references on the topic.

Results

See figure-by-figure comments

Discussion

HMGB1's propensity to phase separate would be helpful if discussed also in the introduction.

We thank the reviewer for the suggestions, however, as phase separation is not the main topic of this work, we felt that it might be more appropriate to refer to HMGB1's propensity to phase separate in the discussion section, rather than in the introduction. We expanded this concept in the discussion.

Add more references on the importance of fuzzy interactions on the formation of condensates and discrete complexes (e.g PMID: 36416859).

We thank the reviewer for the suggestion, we have now added more references on this topic including the aforementioned review and other references recommended by the reviewer.

Discuss in more details previous NMR studies on this complex. Previous work often required looking at other citations to understand the current work.

We have clarified in the text which kind of NMR experiments were previously performed to characterize the interaction and to generate a first 3D model of the complex (page 14).

Discuss in more details the therapeutic value of targeting the heterocomplex interactions.

At the end of the discussion we have added more details regarding the therapeutic value of targeting heterocomplexes.

Reviewer #3 (Remarks to the Author):

In their manuscript, authors studied in detail the molecular interaction (by NMR, ITC and MST) between the chemokine CXCL12 and HMGB1 either with or without its negatively charged intrinsically disordered region (IDR). In addition, they synthesized the IDR domain and evaluated the effect of this domain as such. They prove the importance of the IDR and long-range electrostatic interactions for the low μM affinity heterocomplex interaction between HMGB1 with CXCL12. In contrast to previous reports, authors show that HMGB1 forms an equimolar "fuzzy" 1:1

complex with CXCL12. In addition, they show that acidic IDR of HMGB1 facilitates the binding of the heterocomplex to the chemokine receptor CXCR4.

Comments:

1. Is the effect of the IDR comparable to the effect of negatively charged glycosaminoglycans such as heparan sulfate? Does addition of heparan sulfate have the same effect as addition of the IDR?

The interaction with Ac-pep and with frHMGB1 is in part reminiscent of the interaction with heparin oligosaccharides described by Ziarek et al (2013). As such, the acidic IDR (alone or within frHMGB1), in analogy to negatively charged glycosaminoglycans (GAG), induces significant chemical shift perturbations of the amide resonances of residues located on CXCL12 dimerization surface.

While GAG promote oligomerization of CXCL12, the interaction with frHMGB1, as indicated by AUC experiments, does not promote the oligomerization of CXCL12, as the stoichiometry is 1:1. Along the same line, NMR titrations of CXCL12-LM (i.e. CXCL12 locked in a monomeric conformation) with Ac-pep and frHMGB1 indicate that chemical shift perturbations affecting $\beta 1$ and $\alpha 1$ are ascribable to direct binding, thus $\beta 1$ and $\alpha 1$ constitute a real interaction surface for the acidic IDR.

Is HMGB1 or its IDR releasing CXCL12 from GAGs improving its interaction with CXCR4?

Ziarek et al (2013) have shown by SPR that in vitro the affinity of heparin to CXCL12 is in the nanomolar range (30 nM). The affinity of Ac-pep (0.2 μ M) or of frHMGB1 (2 μ M) appear more than one order of magnitude weaker with respect to heparin, thus we posit that at least in vitro neither Ac-tail nor frHMGB1 might be able to displace heparin from CXCL12. We cannot exclude that in a cellular context the affinity of CXCL12 to its partners might be different and that competition between the CXCL12 interactors might occur. In this context the hypothesis that HMGB1 might release CXCL12 from GAGs improving CXCL12 interaction with CXCR4 is fascinating and worth being investigated in future experiments.

Does addition of heparan sulfate have the same effect as addition of the IDR? Same residues are affected, probably similar surface? (JBC2013)

As stated before, the CXCL12 residues experiencing chemical shift perturbations (CSPs) upon addition of frHMGB1 (and of Ac-pep) in part coincide with the one induced by heparin oligosaccharides and involve residues located on the CXCL12 dimerization surface (residues 23-25, 66, 67). We have added a sentence and the reference to Ziarek et al. 2013 in the discussion highlighting this analogy (page 14). In this context it is important to note that in Ziarek et al the authors used a mutant of CXCL12 locked in a dimeric form, (CXCL12)₂, to eliminate contributions from CXCL12 self-association and define chemical shift perturbations unambiguously caused by heparin binding to the dimeric form of CXCL12. In this case CSPs suggest that heparin oligosaccharides appear to bind orthogonally to the dimerization surface of (CXCL12)₂. In our case, NMR titrations of the locked monomeric form of CXCL12 (CXCL12-LM) with Ac-pep and frHMGB1 have a similar CSP profile as the wild-type protein, thus indicating that $\beta 1$ and $\alpha 1$ constitute a real interaction surface for multivalent interactions with the acidic IDR.

2. Reduced and oxidized forms of HMGB1 have a different activity. Can authors indicate which form of HMGB1 is used in this study and is reduction/oxidation affecting the interactions and the effects of the IDR?

We thank the reviewer for the comment and apologize for the lack of clarity. The main objective of our study was the characterization of the interaction of the fully reduced form of HMGB1 (frHMGB1) with CXCL12. We have now clarified this aspect throughout the manuscript. Prompted by the reviewer's question, for comparison we have also monitored by NMR and MST the binding of oxidized HMGB1 (dsHMGB1) to CXCL12. Our results show that oxidation of HMGB1 reduces but does not totally abolish the interaction with CXCL12 (see response to reviewer 1), in fact, while the interaction with BoxA is reduced upon oxidation, the acidic IDR is still able to interact with CXCL12 (**Supplementary Figure S4**).

References

1. Chu, W.-M., Ivanov, S., Dragoi, A.-M. & Wang, X. A novel role for HMGB1 in TLR9-mediated inflammatory responses to CpG-DNA (89.24). *J. Immunol.* **178**, S153–S153 (2007).
2. Erlendsson, S. & Teilum, K. Binding Revisited—Avidity in Cellular Function and Signaling. *Frontiers in Molecular Biosciences* **7**, (2021).
3. Mittag, T. *et al.* Dynamic equilibrium engagement of a polyvalent ligand with a single-site receptor. *Proc. Natl. Acad. Sci. U. S. A.* **105**, (2008).
4. Eckardt, V. *et al.* Chemokines and galectins form heterodimers to modulate inflammation. *EMBO Rep.* **21**, e47852 (2020).
5. von Hundelshausen, P. *et al.* Chemokine interactome mapping enables tailored intervention in acute and chronic inflammation. *Sci. Transl. Med.* **9**, :eaah6650 (2017).
6. Du, Y. *et al.* Chemokines form nanoparticles with DNA and can superinduce TLR-driven immune inflammation. *J. Exp. Med.* **219**, (2022).
7. Ziarek, J. J. *et al.* Structural basis for chemokine recognition by a G protein–coupled receptor and implications for receptor activation. *Sci. Signal.* **10**, 737–46 (2017).
8. Tompa, P. & Fuxreiter, M. Fuzzy complexes: polymorphism and structural disorder in protein-protein interactions. *Trends Biochem. Sci.* **33**, 2–8 (2008).
9. Wang, X. *et al.* Dynamic Autoinhibition of the HMGB1 Protein via Electrostatic Fuzzy Interactions of Intrinsically Disordered Regions. *J. Mol. Biol.* **433**, 167122 (2021).
10. Manalastas-Cantos, K. *et al.* ATSAS 3.0: Expanded functionality and new tools for small-angle scattering data analysis. *J. Appl. Crystallogr.* **54**, 343–355 (2021).
11. Ngo, T. *et al.* Crosslinking-guided geometry of a complete CXC receptor-chemokine complex and the basis of chemokine subfamily selectivity. *PLoS Biol.* **18**, e3000656. (2020).

REVIEWERS' COMMENTS

Reviewer #1 (Remarks to the Author):

In this revision, the authors have appropriately addressed the issues I raised for the initial manuscript. I do not have any other concerns.

Reviewer #2 (Remarks to the Author):

"The acidic intrinsically disordered region of the inflammatory mediator HMGB1 mediates fuzzy interactions with chemokine CXCL12" – Mantonico et al.

After reading the revised version of this manuscript, I stand by my original assessment that the diversity and depth of structural and biochemical techniques used by the authors to dissect the CXCL12/HMGB1 system is truly impressive. Given the work put into the revisions, the manuscript is now even more complete. I have no hesitation in saying that it is ready for publication.

My major concern with the initial version was that it was not clear that the bar for "fuzzy complex" had been met. The standard for a fuzzy complex is not simply that one or both components of the complex maintain disorder but that the complex itself is disordered. To this end, I cited two examples where the complex is disordered due to the affinity being driven by engagement and exchange between multiple weak motifs (Sic1) and where there is no discernable ordering in the complex (proTa and H1). On first reading of this manuscript, it seemed that the CXCL12/HMGB1 complex was defined by a single micromolar affinity motif. I appreciate that the authors have taken our suggestion and clearly outlined in the introduction how they are defining a fuzzy complex and how they believe CXCL12/HMGB1 fits this model. Further, the clear discussion of dynamics and the addition of relaxation experiments help this narrative greatly.

I also appreciate the work put into addressing my questions with the SAXS data. It is very encouraging to see that the results from Oligomer agree so well with the previously stated model. I wonder why the authors included this (as they mentioned tedious) analysis only for my sake and not as a supplementary figure?

I am quite satisfied with the authors' responses to our review and will be happy to see it published.